# Initialization and Regularization of Factorized Neural Layers

**Mikhail Khodak**
Carnegie Mellon University
khodak@cmu.edu

**Neil Tenenholtz, Lester Mackey, Nicolò Fusi**
Microsoft Research
{netenenh,lmackey,fusi}@microsoft.com

## Abstract

Factorized layers—operations parameterized by products of two or more matrices—occur in a variety of deep learning contexts, including compressed model training, certain types of knowledge distillation, and multi-head self-attention architectures. We study how to initialize and regularize deep nets containing such layers, examining two simple, understudied schemes, *spectral initialization* and *Frobenius decay*, for improving their performance. The guiding insight is to design optimization routines for these networks that are as close as possible to that of their well-tuned, non-decomposed counterparts; we back this intuition with an analysis of how the initialization and regularization schemes impact training with gradient descent, drawing on modern attempts to understand the interplay of weight-decay and batch-normalization. Empirically, we highlight the benefits of spectral initialization and Frobenius decay across a variety of settings. In model compression, we show that they enable low-rank methods to significantly outperform both unstructured sparsity and tensor methods on the task of training low-memory residual networks; analogs of the schemes also improve the performance of tensor decomposition techniques. For knowledge distillation, Frobenius decay enables a simple, *overcomplete* baseline that yields a compact model from over-parameterized training without requiring retraining with or pruning a teacher network. Finally, we show how both schemes applied to multi-head attention lead to improved performance on both translation and unsupervised pre-training.

## 1 Introduction

Most neural network layers consist of matrix-parameterized functions followed by simple operations such as activation or normalization. These layers are the main sources of model expressivity, but also the biggest contributors to computation and memory cost; thus modifying these layers to improve computational performance while maintaining performance is highly desirable. We study the approach of *factorizing* layers, i.e. reparameterizing them so that their weights are defined as products of two or more matrices. When these are smaller than the original matrix, the resulting networks are more efficient for both training and inference (Denil et al., 2013; Moczulski et al., 2015; Ioannou et al., 2016; Tai et al., 2016), resulting in *model compression*. On the other hand, if training cost is not a concern, one can increase the width or depth of the factors to over-parameterize models (Guo et al., 2020; Cao et al., 2020), improving learning without increasing inference-time cost. This can be seen as a simple, teacher-free form of knowledge distillation. Factorized layers also arise implicitly, such as in the case of multi-head attention (MHA) (Vaswani et al., 2017).

Despite such appealing properties, networks with factorized neural layers are non-trivial to train from scratch, requiring custom initialization, regularization, and optimization schemes. In this paper we focus on initialization, regularization, and how they interact with gradient-based optimization of factorized layers. We first study *spectral initialization* (SI), which initializes factors using singular value decomposition (SVD) so that their product approximates the target un-factorized matrix. Then, we study *Frobenius decay* (FD), which regularizes the product of matrices in a factorized layer rather than its individual terms. Both are motivated by matching the training regimen of the analogous un-factorized optimization. Note that SI has been previously considered in the context of model compression, albeit usually for factorizing pre-trained models (Nakkiran et al., 2015; Yaguchi et al., 2019; Yang et al., 2020) rather than low-rank initialization for end-to-end training; FD has been used in model compression using an uncompressed teacher (Idelbayev & Carreira-Perpiñán, 2020).

We formalize and study the justifications of SI and FD from both the classical perspective—matching the un-factorized objective and scaling—and in the presence of BatchNorm (Ioffe & Szegedy, 2015), where this does not apply. Extending recent studies of weight-decay (Zhang et al., 2019), we argue that the effective step-size at spectral initialization is controlled by the factorization's Frobenius norm and show convincing evidence that weight-decay penalizes the *nuclear* norm.

We then turn to applications, starting with low-memory training, which is dominated by unstructured sparsity methods—i.e. guessing "lottery tickets" (Frankle & Carbin, 2019)—with a prevailing trend of viewing low-rank methods as uncompetitive for compression (Blalock et al., 2020; Zhang et al., 2020; Idelbayev & Carreira-Perpiñán, 2020; Su et al., 2020). Here we show that, without tuning, factorized neural layers outperform all structured sparsity methods on ResNet architectures (He et al., 2016), despite lagging on VGG (Simonyan & Zisserman, 2015). Through ablations, we show that this result is due to using both SI and FD on the factorized layers. We further compare to a recent evaluation of tensor-decomposition approaches for compressed WideResNet training (Zagoruyko & Komodakis, 2016; Gray et al., 2019), showing that (a) low-rank approaches with SI and FD can outperform them and (b) they are themselves helped by tensor-variants of SI and FD.

We also study a fledgling subfield we term *overcomplete* knowledge distillation (Arora et al., 2018; Guo et al., 2020; Cao et al., 2020) in which model weights are over-parameterized as overcomplete factorizations; after training the factors are multiplied to obtain a compact representation of the same network. We show that FD leads to significant improvements, e.g. we outperform ResNet110 with an overcomplete ResNet56 that takes 1.5x less time to train and has 2x fewer parameters at test-time.

Finally, we study Transformer architectures, starting by showing that FD improves translation performance when applied to MHA. We also show that SI is critical for low-rank training of the model's linear layers. In an application to BERT pre-training (Devlin et al., 2019), we construct a Frobenius-regularized variant—*FLAMBé*—of the LAMB method (You et al., 2020), and show that, much like for transformers, it improves performance both for full-rank and low-rank MHA layers.

To summarize, our main contributions are (1) motivating the study of training factorized layers via both the usual setting (model compression) and recent applications (distillation, multi-head attention), (2) justifying the use of SI and FD mathematically and experimentally, and (3) demonstrating their effectiveness by providing strong baselines and novel advances in many settings. Code to reproduce our results is available here: `https://github.com/microsoft/fnl_paper`.

## 1.1 RELATED WORK

We are not the first study gradient descent on factorized layers; in particular, deep linear nets are well-studied in theory (Saxe et al., 2014; Gunasekar et al., 2019). Apart from Bernacchia et al. (2018) these largely examine existing algorithms, although Arora et al. (2018) do effectively propose overcomplete knowledge distillation. Rather than the descent method, we focus on the initialization and regularization. For the former, several papers use SI *after* training (Nakkiran et al., 2015; Yaguchi et al., 2019; Yang et al., 2020), while Ioannou et al. (2016) argue for initializing factors as though they were single layers, which we find inferior to SI in some cases. Outside deep learning, spectral methods have also been shown to yield better initializations for certain matrix and tensor problems (Keshavan et al., 2010; Chi et al., 2019; Cai et al., 2019). For regularization, Gray et al. (2019) suggest *compression-rate scaling* (CRS), which scales weight-decay using the reduction in parameter count; this is justified via the usual Bayesian understanding of $\ell_2$-regularization (Murphy, 2012). However, we find that FD is superior to any tuning of regular weight-decay, which subsumes CRS. Our own analysis is based on recent work suggesting that the function of weight-decay is to aid optimization by preventing the effective step-size from becoming too small (Zhang et al., 2019).

## 2 PRELIMINARIES ON FACTORIZED NEURAL LAYERS

In the training phase of (self-)supervised ML, we often solve optimization problems of the form $\min_{\theta \in \Theta} \frac{1}{|S|} \sum_{(\boldsymbol{x}, \boldsymbol{y}) \in S} \ell(f_\theta(\boldsymbol{x}), \boldsymbol{y}) + \Omega(\theta)$, where $f_\theta : \mathcal{X} \mapsto \mathcal{Y}$ is a function from input domain $\mathcal{X}$ to output domain $\mathcal{Y}$ parameterized by elements $\theta \in \Theta$, $\ell : \mathcal{Y} \times \mathcal{Y} \mapsto \mathbb{R}$ is a scalar-valued loss function, $\Omega : \Theta \mapsto \mathbb{R}$ is a scalar-valued regularizer, and $S \subset \mathcal{X} \times \mathcal{Y}$ is a finite set of (self-)supervised training examples. We study the setting where $f_\theta$ is a neural network, an $L$-layer function whose parameters $\theta$ consist of $L$ matrices $\boldsymbol{W}_i \in \mathbb{R}^{m_i \times n_i}$ and whose output $f_\theta(\boldsymbol{x})$ given input $\boldsymbol{x}$ is defined recursively using $L$ functions $g_i$ via the formula $\boldsymbol{x}_i = g_i(\boldsymbol{W}_i, \boldsymbol{x}_{i-1})$, with $\boldsymbol{x}_0 = \boldsymbol{x}$ and $f_\theta(\boldsymbol{x}) = \boldsymbol{x}_L$.

The standard approach to training $f_\theta$ is to specify the regularizer $\Omega$, (randomly) pick an initialization in $\Theta$, and iteratively update the parameters using some first-order algorithm such as SGD to optimize the objective above until some stopping criterion is met. However, in many cases we instead optimize over factorized variants of these networks, in which some or all of the matrices $W_i \in \mathbb{R}^{m_i \times n_i}$ are re-parameterized as a product $W_i = U_i(\prod_{j=1}^{d_i} M_{ij})V_i^T$ for some inner depth $d_i \geqslant 0$ and matrices $U_i \in \mathbb{R}^{m_i \times r_i}, V_i \in \mathbb{R}^{n_i \times r_i}$, and $M_{ij} \in \mathbb{R}^{r_i \times r_i} \ \forall \ j$. As discussed in the following examples, this can be done to obtain better generalization, improve optimization, or satisfy practical computational or memory constraints during training or inference. For simplicity, we drop the subscript $i$ whenever re-parameterizing only one layer and only consider the cases when inner depth $d$ is 0 or 1.

## 2.1 FULLY-CONNECTED LAYERS

A fully-connected layer takes an $n$-dimensional input $x_{i-1}$ and outputs an $m$-dimensional vector $x_i = \sigma(W x_{i-1})$, where $\sigma : \mathbb{R}^m \mapsto \mathbb{R}^m$ is an element-wise activation function. Here, decomposing $W \in \mathbb{R}^{m \times n}$ into the product $UV^T$, where $U \in \mathbb{R}^{m \times r}, V \in \mathbb{R}^{n \times r}$, and setting $r \ll \min\{m, n\}$ reduces computation and memory costs from $\mathcal{O}(mn)$ to $\mathcal{O}(mr + nr)$. We refer to this setting as *model compression*. Standard learning theory suggests that a small rank $r$ also improves generalization, e.g. for a factorized fully-connected ReLU network, applying $\|W\|_F^2 / \|W\|_2^2 \leqslant \mathrm{rank}(W)$ to Neyshabur et al. (2018, Theorem 1) and substituting $W_i = U_i V_i^T$ gives a w.h.p. margin-bound $\tilde{\mathcal{O}}(\sqrt{mr/|\mathcal{S}|})$ suggesting that generalization error varies with the square root of the rank (see Corollary A.1).

Alternatively, by setting $r \geqslant \min\{m, n\}$ and/or including an inner matrix $M \in \mathbb{R}^{r \times r}$, we can attempt to take advantage of improved optimization due to increased width (Du & Hu, 2019) and/or increased depth (Arora et al., 2018). Crucially, this does not increase inference costs because we can recompose the matrix after training and just use the product. As the goal is to obtain a better small model by first training a large one, we refer to this setting as *overcomplete knowledge distillation*; of course, unlike regular distillation it is much simpler since there is no student-teacher training stage.

## 2.2 CONVOLUTIONAL LAYERS

A 2d convolutional layer takes an $h \times w \times c_{i-1}$-dimensional input $x_{i-1}$ and outputs a $h \times w \times c_i$-dimensional output $x_i$ defined by convolving $c_i$ different $k \times k$ filters over each of $c_{i-1}$ input channels. Often the result is passed through a nonlinearity. 2d convolutional layers are parameterized by $c_i \times c_{i-1} \times k \times k$ tensors and require $O(k^2 c_i c_{i-1})$ memory and compute. A straightforward way of factorizing this tensor without using tensor decomposition is to reshape it into a $c_i k \times c_{i-1} k$ matrix $W$, which can then be decomposed as $W = UV^T$ for $U \in \mathbb{R}^{c_i k \times r}, V \in \mathbb{R}^{c_{i-1} k \times r}$ and some rank $r > 0$. As in the fully-connected case, we can either set the rank $r$ to be small in order to reduce the number of parameters or alternatively increase the width ($r$) or the depth ($d$) of the factorization to do overcomplete knowledge distillation.

Note that in the low-rank case a naive approach does not save computation since we must first multiply $U$ and $V^T$, reshape the product $UV^T$, and then use the resulting tensor in a regular 2d convolution of the original size and complexity. However, as shown by Tai et al. (2016), applying the 2d $k \times k$ convolution with $c_{i-1}$ input channels and $c_i$ output channels obtained by reshaping $UV^T$ is equivalent to a composition of two 1d convolutions: the first defined by $V^T \in \mathbb{R}^{r \times c_{i-1} k}$ consists of $r$ output channels and filters of size $k$ along one input dimension and the second defined by $U \in \mathbb{R}^{c_i k \times r}$ consists of $c_i$ output channels and filters of size $k$ along the other input dimension. Together the two 1d convolutions require $O(kr(c_i + c_{i-1}))$ memory and computation, which is significantly better than the $O(k^2 c_i c_{i-1})$ cost of the unfactorized case if $r \ll k \min\{c_i, c_{i-1}\}$.

## 2.3 MULTI-HEAD ATTENTION

An MHA layer (Vaswani et al., 2017) with $H$ attention heads and hidden dimension $d$ can be expressed as being parameterized with $4H$ matrices: one each of $Q_h, K_h, V_h, O_h \in \mathbb{R}^{d \times d/H}$ for each head $h$. Then for a length-$T$ input $x \in \mathbb{R}^{T \times d}$ it outputs

$$\sum_{h=1}^{H} \mathrm{Softmax}\left(\frac{x Q_h K_h^T x^T}{\sqrt{d/H}}\right) x V_h O_h^T \tag{1}$$

MHA combines $2H$ quadratic forms $Q_h K_h^T, V_h O_h^T$ of rank $r = d/H$, each a product of matrices, i.e. a factorized layer. We refer to the first form as "Query-Key" and the second as "Output-Value." Note that $r$ can be varied independently of $d$ to change expressivity, memory, and computation.

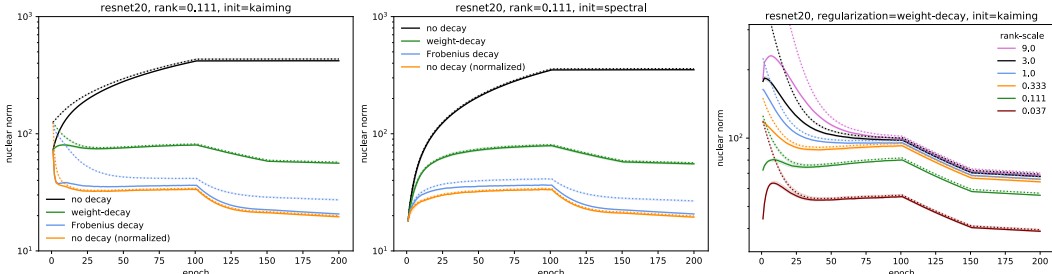

Figure 1: Average nuclear norm across factorized layers of ResNet20 during CIFAR-10 training when initialized regularly (left) and using SI (center); the dotted line is the upper bound on the nuclear norm regularized by weight-decay (2). The right plot track the same quantities for the case of regular weight-decay across different rank-scales. "no decay (normalized)" normalizes matrix factors after each step to have the same norm as "Frobenius decay " (detailed in Section 3.3).

## 3 INITIALIZATION AND REGULARIZATION

We now define the initialization and regularization schemes we study as natural extensions of techniques for non-factorized models. They thus require no tuning when an existing training implementation of a non-factorized deep net is available. We later discuss how to justify these schemes when layers are normalized, e.g. using BatchNorm (Ioffe & Szegedy, 2015). In all experiments with convolutional models in this and subsequent sections we factorize all layers except the first and last, which are small, and determine layer ranks by multiplying a uniform scaling factor by the product of a layer's output channels and kernel width. This *rank-scale* can be varied to attain the desired number of parameters. Note that our approach may be further improved via a more sophisticated or adaptive rank-assignment scheme (Idelbayev & Carreira-Perpiñán, 2020).

### 3.1 SPECTRAL INITIALIZATION

Initialization is a major focus of deep learning research (He et al., 2015; Mishkin & Matas, 2016; Yang & Schoenholz, 2017). A common approach is to prevent compounding changes in the norms of the intermediate representations across layers caused by repeated multiplication by the weight matrices. The *spectral initialization* scheme for initializing *low-rank* factorized layers attempts to inherit, when possible, this property from an existing initialization by using SVD to ensure that the resulting product matrix is as close as possible to the original parameter:

**Definition 3.1.** *Let $\boldsymbol{W} \in \mathbb{R}^{m \times n}$ be a parameter of an unfactorized layer. For $r \leqslant \min\{m, n\}$ the* **spectral initialization** *(SI) of the factors $\boldsymbol{U} \in \mathbb{R}^{m \times r}, \boldsymbol{V} \in \mathbb{R}^{n \times r}$ of the corresponding factorized layer sets $\boldsymbol{U} = \tilde{\boldsymbol{U}}\sqrt{\Sigma}$ and $\boldsymbol{V} = \tilde{\boldsymbol{V}}\sqrt{\Sigma}$, for $\tilde{\boldsymbol{U}}, \Sigma, \tilde{\boldsymbol{V}} = \mathrm{SVD}_r(\boldsymbol{W})$ given by the rank-$r$ SVD of $\boldsymbol{W}$.*

SI preserves the largest singular value of $\boldsymbol{W}$, so if the original scheme did not suffer from a compounding increase in representation norm then neither will spectral initialization. On the other hand, while low-rank layers impose a nullspace, SI aligns it with the directions minimized by $\boldsymbol{W}$.

### 3.2 FROBENIUS DECAY

Weight-decay is a common regularizer for deep nets, often implemented explicitly by adding $\Omega(\theta) = \frac{\lambda}{2}\sum_{\boldsymbol{W} \in \theta} \|\boldsymbol{W}\|_F^2$ to the objective for some $\lambda \geqslant 0$. Classically, it is thought to improve generalization by constraining model capacity. When training factorized layers parameterized by $\boldsymbol{U}(\prod_{j=1}^{d} \boldsymbol{M}_j)\boldsymbol{V}^T$, the easiest approach to implement is to replace each $\frac{\lambda}{2}\|\boldsymbol{W}\|_F^2$ term in $\Omega(\theta)$ by $\frac{\lambda}{2}\left(\|\boldsymbol{U}\|_F^2 + \|\boldsymbol{V}\|_F^2 + \sum_{j=1}^{d}\|\boldsymbol{M}_j\|_F^2\right)$. However, this yields a very different optimization problem: for example, if we consider the case of $d = 0$ then this regularizer is in fact an upper bound on the nuclear norm of the recomposed matrix $\boldsymbol{U}\boldsymbol{V}^T$ (Srebro & Shraibman, 2005, Lemma 1):

$$\frac{\lambda}{2}\left(\|\boldsymbol{U}\|_F^2 + \|\boldsymbol{V}\|_F^2\right) \geqslant \min_{\tilde{\boldsymbol{U}}\tilde{\boldsymbol{V}}^T=\boldsymbol{U}\boldsymbol{V}^T} \frac{\lambda}{2}\left(\|\tilde{\boldsymbol{U}}\|_F^2 + \|\tilde{\boldsymbol{V}}\|_F^2\right) = \lambda\|\boldsymbol{U}\boldsymbol{V}^T\|_* \tag{2}$$

In fact, Figure 1 shows that for weight-decay this upper bound is tight throughout the training of factorized ResNet20 across a variety of ranks and initializations, suggesting that the naive approach is indeed regularizing the nuclear norm rather than the Frobenius norm.

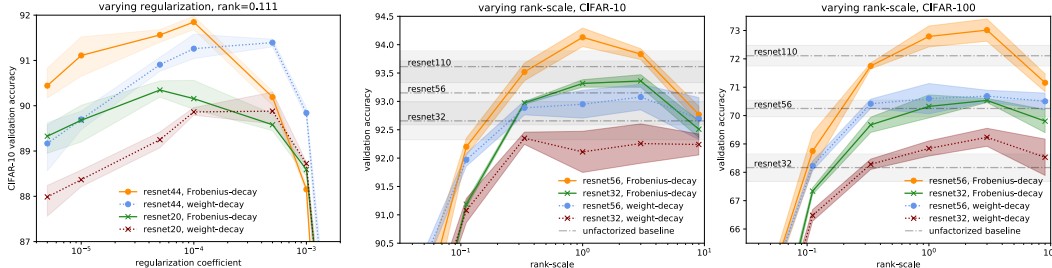

Figure 2: Comparison of weight-decay and FD at different regularization levels (left) and different rank-scales (center and right) when training factorized ResNets on CIFAR.

Since compression already constrains capacity, in the low-rank case one might favor just reducing regularization, e.g. multiplying $\lambda$ by the compression rate (Gray et al., 2019). However, Figure 2 shows that this can lead to *worse* performance, and the approach still penalizes the nuclear norm. *Frobenius decay* avoids this issue by simply penalizing the squared norm of the entire factorization:

**Definition 3.2.** *For $\lambda \geqslant 0$ let $\frac{\lambda}{2}\|\boldsymbol{W}\|_F^2$ be the contribution of an unfactorized layer parameterized by $\boldsymbol{W} \in \mathbb{R}^{m \times n}$ to the penalty term. Then the* **Frobenius decay** *(FD) penalty on matrices $\boldsymbol{U} \in \mathbb{R}^{m \times r}$, $\boldsymbol{V} \in \mathbb{R}^{n \times r}$, and $\boldsymbol{M}_j \in \mathbb{R}^{r \times r}$ of the corresponding factorized layer is $\frac{\lambda}{2} \left\| \boldsymbol{U}(\prod_{j=1}^d \boldsymbol{M}_j)\boldsymbol{V}^T \right\|_F^2$.*

By substituting the factorization directly, FD makes the least change to the problem: rank-$r$ optima of the non-factorized objective will also minimize the factorized one. We can also bound generalization error of the ReLU net from Section 2.1 by a term $\tilde{\mathcal{O}}\left(\sqrt{\frac{m}{|\mathcal{S}|} \sum_{i=1}^L \|\boldsymbol{U}_i(\prod_{j=1}^d \boldsymbol{M}_{ij})\boldsymbol{V}_i^T\|_F^2}\right)$ varying directly with the quantity penalized by FD (see Corollary A.2). Notably, FD is a *stronger* penalty than the nuclear norm implicitly regularized by weight-decay, yet it still yields better models.

### 3.3 Initialization and Regularization in the Presence of Normalization

The use of spectral initialization and Frobenius decay is largely motivated by the norms of the recomposed matrices: SI prevents them increasing feature vector norms across layers, while FD constrains model capacity via parameter norms. However, normalization layers like BatchNorm (Ioffe & Szegedy, 2015) and others (Ba et al., 2016) largely negate the forward-pass and model-capacity effects of the norms of the weights parameterizing the layers they follow. Thus for most modern models we need a different explanation for the effectiveness of SI and FD.

Despite the fact that most layers' norms do not affect inference or capacity, weight-decay remains useful for optimizing deep nets. Recently, Zhang et al. (2019) extended the analysis of Hoffer et al. (2018) to argue that the *effective step-size* of the weight *direction* $\hat{\boldsymbol{W}}$ is roughly $\eta/\|\boldsymbol{W}\|_F^2$, where $\eta$ is the SGD step-size. Thus by preventing the norm of $\boldsymbol{W}$ from growing too large weight-decay maintains a large-enough effective step-size during training. We draw on this analysis to explore the effect of SI and FD on factorized models. For simplicity, we define a normalized layer to be one that does not depend on the scale of its parameter. Ignoring stability offset terms, this definition roughly holds for normalized linear layers, convolutional layers in ResNets, and the Output-Value quadratic form in Transformers if the residual connection is added after rather than before normalization.

**Definition 3.3.** *A normalized layer $g(\boldsymbol{W}, \boldsymbol{x})$ parameterized by $\boldsymbol{W} \in \mathbb{R}^{m \times n}$ is one that satisfies $g(\boldsymbol{W}, \boldsymbol{x}) = g(\rho \boldsymbol{W}, \boldsymbol{x})$ for all $\boldsymbol{W} \in \mathbb{R}^{m \times n}$ and all positive scalars $\rho$.*

Because the output does not depend on the magnitude of $\boldsymbol{U}\boldsymbol{V}^T$, what matters is the direction of the composed matrix. During an SGD step this direction is updated as follows (proof in Appendix B):

**Claim 3.1.** *At all steps $t \geqslant 0$ let $g$ be a normalized layer of a differentiable model $f_{\theta_t} : \mathcal{X} \mapsto \mathcal{Y}$ parameterized by $\boldsymbol{U}_t\boldsymbol{V}_t^T$ for $\boldsymbol{U}_t \in \mathbb{R}^{m \times r}, \boldsymbol{V}_t^T \in \mathbb{R}^{n \times r}, r \geqslant 1$. Suppose we update $\boldsymbol{P} = \boldsymbol{U}$ and $\boldsymbol{P} = \boldsymbol{V}$ by SGD, setting $\boldsymbol{P}_{t+1} \leftarrow \boldsymbol{P}_t - \eta \nabla_{\boldsymbol{P}_t}$ using a gradient $\nabla_{\boldsymbol{P}_t} = \frac{1}{|\mathcal{B}|} \sum_{(\boldsymbol{x},\boldsymbol{y}) \in \mathcal{B}} \nabla_{\boldsymbol{P}_t} \ell(f_{\theta_t}(\boldsymbol{x}), \boldsymbol{y})$ over batch $\mathcal{B} \subset (\mathcal{X}, \mathcal{Y})$ and $\eta > 0$ sufficiently small. Then for $\hat{\nabla}_t = \nabla_{\hat{\boldsymbol{W}}_t} \boldsymbol{V}_t \boldsymbol{V}_t^T + \boldsymbol{U}_t \boldsymbol{U}_t^T \nabla_{\hat{\boldsymbol{W}}_t}$ we have that the vectorized direction $\hat{\boldsymbol{w}}_t = \text{vec}(\hat{\boldsymbol{W}}_t)$ of $\boldsymbol{W}_t = \boldsymbol{U}_t\boldsymbol{V}_t^T$ is updated as*

$$\hat{\boldsymbol{w}}_{t+1} \leftarrow \hat{\boldsymbol{w}}_t - \frac{\eta}{\|\boldsymbol{W}_t\|_F^2} \left(\boldsymbol{I}_{mn} - \hat{\boldsymbol{w}}_t \hat{\boldsymbol{w}}_t^T\right) \text{vec}(\hat{\nabla}_t) + \mathcal{O}(\eta^2) \tag{3}$$

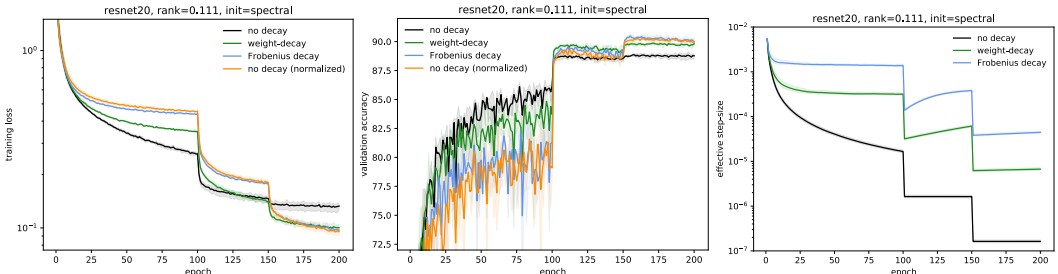

Figure 3: Traces depicting training of low-rank ResNet-20 with different decay settings; "no decay (normalized)" normalizes matrix factors after each step to have the same norm as "Frobenius decay." The effective step size (right) is the average over convolution layers $i$ of $\eta/\|\boldsymbol{U}_i\boldsymbol{V}_i^T\|_F^2$.

Note that (1) we ignore decay because $\lambda = \mathcal{O}(\eta)$ so any resulting term is $\mathcal{O}(\eta^2)$ and (2) the update rule is almost the same as that obtained for the unfactorized case by Zhang et al. (2019), except they have $\hat{\nabla}_t$ as the true gradient of the direction. Thus, apart from a rank-one correction, $\hat{\boldsymbol{W}}_t$ is approximately updated with step-size $\eta/\|\boldsymbol{W}_t\|_F^2$ multiplying a linear transformation of its gradient. To understand the nature of this transformation, note that at spectral initialization we have that $\boldsymbol{V}_0\boldsymbol{V}_0^T = \boldsymbol{U}_0\boldsymbol{U}_0^T = \Sigma_r$ are diagonal matrices of singular values of the full-rank initialization $\boldsymbol{W}$; furthermore, if $\boldsymbol{W}$ is a Gaussian ensemble with scale $1/\sqrt{n}$, which roughly aligns with common initialization schemes (He et al., 2015), then its singular values are roughly distributed around 1 and supported on $[0, 2]$ (Bai & Yin, 1993). Since $\hat{\nabla}_t = \nabla_{\hat{\boldsymbol{W}}_t}\boldsymbol{V}_t\boldsymbol{V}_t^T + \boldsymbol{U}_t\boldsymbol{U}_t^T\nabla_{\hat{\boldsymbol{W}}_t}$, this suggests that, at spectral initialization, an effective learning rate of $\eta/\|\boldsymbol{W}_0\|_F^2$ is a reasonable approximation for the factorized update. This points to the role of SI being to initialize the factorization at an appropriate scale and perhaps also to make the first update more aligned with the gradient w.r.t. $\hat{\boldsymbol{W}}_0$.

As in the unfactorized case, our analysis suggests that, the main role of decay may be to maintain a large effective learning rate $\eta/\|\boldsymbol{W}\|_F^2$; furthermore, FD may be more effective than regular decay because it provides stronger regularization and directly penalizes the quantity of interest. We support this hypothesis using experiments analogous to Zhang et al. (2019) by comparing training a low-rank ResNet-20 with FD to training it with no decay but at each step normalizing all BatchNormed layers to have the same Frobenius norm as the FD run. In Figure 3 we see that the latter scheme closely tracks the former in terms of both the training loss and the test accuracy. Figure 3 also shows that FD maintains a higher effective step-size than regular weight-decay throughout training.

## 4 COMPRESSED MODEL TRAINING: LOW-RANK, SPARSE, AND TENSORIAL

We first study SI and FD for training factorized models with low-rank convolutional layers, comparing against the dominant approaches to low-memory training: sparse layers and tensor decomposition. For direct comparison we evaluate models with near-identical parameter counts; note that by normalizing by memory we *disadvantage* the low-rank approach, which often has a comparative advantage for speed. All models are trained for 200 epochs with the same optimizer settings as for the unfactorized models; the weight-decay coefficient is left unchanged when replacing by FD.

**Low-Rank and Sparse Training:** We train modified ResNet32 and VGG19 used in the lottery-ticket-guessing literature (Wang et al., 2020; Su et al., 2020). Motivated by Frankle & Carbin (2019), such methods fix a sparsity pattern at initialization and train only the unpruned weights. While they achieve high parameter savings, their computational advantages over full models are less clear, as software and accelerators often do not efficiently implement arbitrary sparsity (Paszke et al., 2019; NVIDIA, 2020). In contrast, we do see acceleration via low-rank convolutions, almost halving the time of ResNet's forward and backward pass at the highest compression. For completeness, we also show methods that vary the sparsity pattern (dynamic), prune trained models (pruning), and prune trained models and retrain (lottery); note that the latter two require training an uncompressed model.

In Table 1 we see that the low-rank approach, with SI & FD, dominates at the higher memory settings of ResNet across all three datasets considered, often outperforming even approaches that train an uncompressed model first. It is also close to the best compressed training approach in the lowest memory setting for CIFAR-100 (Krizhevksy, 2009) and Tiny-ImageNet (Deng et al., 2009).

Table 1: Comparison of low-rank and sparse training in a common evaluation setting for "ticket-guessing" (Wang et al., 2020; Su et al., 2020). Best results overall are *italicized*; best results from full low-memory training are **bolded**. For complete results and deviations see Tables 5 and 6.

| Regime | VGG19[NFC] (full model)[*] Compressed (from ≈20M) | CIFAR-10 (93.75±0.48) 0.10 | 0.05 | 0.02 | CIFAR-100 (73.42±0.17) 0.10 | 0.05 | 0.02 | Tiny-ImageNet (62.19±0.74) 0.10 | 0.05 | 0.02 |
|---|---|---|---|---|---|---|---|---|---|---|
| Regular Training | Pruning[*] | 93.83 | 93.69 | 93.49 | 73.83 | 73.07 | *71.69* | 61.21 | 60.49 | 55.55 |
| | Lottery[†] (w. hybrid/rewind) | *94.14* | *93.99* | *93.52* | *74.06* | *73.10* | 71.61 | 61.19 | *60.57* | 56.18 |
| Low Memory Training | Dynamic Sparsity[‡] | 93.75 | **93.86** | **93.13** | 72.36 | **71.98** | 70.70 | *62.49* | 59.81 | *58.36* |
| | Fixed Sparsity[#] | **93.77** | 93.43 | 92.45 | **72.84** | 71.83 | 68.98 | 61.02 | 59.50 | 57.28 |
| | Low-Rank[*] | 90.71 | 88.08 | 80.81 | 62.32 | 54.97 | 42.80 | 48.50 | 46.98 | 36.57 |
| | Low-Rank (SI)[*] | 90.99 | 88.61 | 82.53 | 63.77 | 58.51 | 45.02 | 49.87 | 46.73 | 37.25 |
| | Low-Rank (FD)[*] | 91.57 | 88.16 | 82.81 | 64.07 | 59.34 | 45.73 | 53.93 | 47.95 | 35.63 |
| | Low-Rank (SI & FD)[*] | 91.58 | 88.98 | 83.27 | 65.61 | 60.1 | 46.43 | 53.53 | 47.60 | 35.68 |
| Regime | ResNet32[x2] (full model)[*] Compressed (from ≈7.5M) | CIFAR-10 (94.49±0.29) 0.10 | 0.05 | 0.02 | CIFAR-100 (75.41±1.26) 0.10 | 0.05 | 0.02 | Tiny-ImageNet (63.02±0.94) 0.15 | 0.10 | 0.05 |
| Regular Training | Pruning[*] | 94.21 | *93.29* | 90.31 | 72.34 | 68.73 | 60.65 | 58.86 | 57.62 | 51.70 |
| | Lottery[†] (w. hybrid/rewind) | 94.14 | *93.02* | *90.85* | 72.41 | 69.28 | *63.44* | 60.31 | 57.77 | 51.21 |
| Low Memory Training | Dynamic Sparsity[‡] | 92.97 | 91.61 | 88.46 | 69.66 | 68.20 | *62.25* | 57.08 | 57.19 | *56.18* |
| | Fixed Sparsity[#] | 92.97 | 91.60 | **89.10** | 69.70 | 66.82 | 60.11 | 57.25 | 55.53 | 51.41 |
| | Low-Rank[*] | 93.59 | 92.45 | 87.95 | 72.71 | 67.86 | 61.08 | 60.82 | 58.72 | 55.39 |
| | Low-Rank (SI)[*] | 92.52 | 91.72 | 87.36 | 71.71 | 67.41 | 60.79 | 59.53 | 57.60 | 55.00 |
| | Low-Rank (FD)[*] | 92.92 | 89.44 | 83.05 | 71.85 | 66.89 | 55.31 | 59.45 | 57.24 | 54.15 |
| | Low-Rank (SI & FD)[*] | *94.34* | **92.90** | 87.97 | *74.41* | *70.22* | 61.40 | *62.24* | *60.25* | 55.97 |

[*] Best of (LeCun et al., 1990; Zeng & Urtasun, 2019), obtained from Wang et al. (2020).
[†] Best of (Frankle & Carbin, 2019; Renda et al., 2020; Su et al., 2020), obtained from Wang et al. (2020) and Su et al. (2020).
[‡] Best of (Mostafa & Wang, 2019; Mocanu et al., 2018; Bellec et al., 2018), obtained from Wang et al. (2020).
[#] Best of (Lee et al., 2018; Wang et al., 2020; Su et al., 2020), obtained from Wang et al. (2020) and Su et al. (2020).
[*] Our method or our reproduction; results averaged over three random trials.

Table 2: Comparison of low-rank and tensor-decomposition training of WideResNet28-10 on CIFAR-10 (mean of 3 trials) with different regularization and initialization. Best errors **bolded**.

| compression | decomposition[*] | WD[†] | WD[†] & SI | CRS[‡] | CRS[‡] & SI | FD | FD & SI |
|---|---|---|---|---|---|---|---|
| none (≈36.5M param.) | | 3.21±0.22 | | | | | |
| 0.01667 (≈0.61M param.) | Low-Rank | 7.62±0.08 | 7.72±0.11 | 8.63±0.87 | 8.83±0.17 | 7.23±0.32 | 7.26±0.45 |
| | Tensor-Train | 8.55±4.22 | 7.72±0.46 | 7.01±0.35 | 6.52±0.17 | 7.26±0.73 | **6.22±0.50** |
| 0.06667 (≈2.4M param.) | Low-Rank | 5.31±0.06 | 5.49±0.42 | 5.86±0.22 | 5.53±0.72 | 3.89±0.17 | **3.86±0.22** |
| | Tensor-Train | 7.33±0.27 | 6.49±0.30 | 5.20±0.37 | 4.79±0.15 | 5.02±0.15 | 5.10±0.35 |

[*] See Table 7 for results using the Tucker decomposition, which generally performs worse than Tensor-Train at the same compression.
[†] Regular weight-decay with $\lambda = 0.005$.
[‡] Regular weight-decay with coefficient scaled by the compression rate (Gray et al., 2019).

On the other hand, the low-rank approach is substantially worse for VGG; nevertheless, ResNet is both smaller and more accurate, so if the goal is an accurate compressed model learned from scratch then one should prefer the low-rank approach. Our results demonstrate a strong, simple baseline not frequently compared to in the low-memory training literature (Frankle & Carbin, 2019; Wang et al., 2020; Su et al., 2020). In fact, since it preserves the top singular components of the original weights, SI can itself be considered a type of (spectral) magnitude pruning. Finally, Table 1 highlights the complementary nature of SI and FD, which outperform regular low-rank training on both models, although interestingly they consistently *decrease* ResNet performance when used separately.

**Matrix and Tensor Decomposition:** We next compare against tensor decompositions, another common approach to small model training (Kossaifi et al., 2020a;b). A recent evaluation of tensors and other related approaches by Gray et al. (2019) found that Tensor-Train decomposition (Oseledets, 2011) obtained the best memory-accuracy trade-off on WideResNet; we thus compare directly to this approach. Note that, while we normalize according to memory, Tensor-Train must be expanded to the full tensor prior to convolution and thus increases the required compute, unlike low-rank factorization. In Table 2 we show that at 6.7% of the original parameters the low-rank approach with FD and SI significantly outperforms Tensor-Train. Tensor-Train excels at the highly compressed 1.7% setting but is greatly improved by leveraging tensor analogs of SI—decomposing a random initialization rather than randomly initializing tensor cores directly—and FD—penalizing the squared Frobenius norm of the full tensor. We also compare to CRS (Gray et al., 2019), which scales regular weight-decay by the compression rate. It is roughly as beneficial as FD across different evaluations of Tensor-Train, but FD is significantly better for low-rank factorized neural layers.

Table 3: Overcomplete ResNet performance (mean of 3 trials). Best at each depth is **bolded**; cases where we match the next deeper network with around the same *training* memory are underlined.

| data | ResNet depth | metric | unfactorized WD | DO-Conv* WD | full WD | full FD | deep WD | deep FD | wide WD | wide FD |
|---|---|---|---|---|---|---|---|---|---|---|
| CIFAR 10 | 32 | test accuracy (%) | 92.66 | N/A | 92.11 | 93.32 | 91.37 | 93.11 | 92.26 | **93.36** |
| | | training param. (M) | 0.46 | N/A | 0.95 | 0.95 | 1.43 | 1.43 | 2.84 | 2.84 |
| | | testing param. (M) | 0.46 | 0.46 | 0.46 | 0.46 | 0.46 | 0.46 | 0.46 | 0.46 |
| | 56 | test accuracy (%) | 93.15 | 93.38 | 92.95 | **94.13** | 92.52 | 93.89 | 93.08 | 93.83 |
| | | training param. (M) | 0.85 | N/A | 1.72 | 1.72 | 2.59 | 2.59 | 5.16 | 5.16 |
| | | testing param. (M) | 0.85 | 0.85 | 0.85 | 0.85 | 0.85 | 0.85 | 0.85 | 0.85 |
| | 110 | test accuracy (%) | 93.61 | 93.93 | 93.73 | 94.28 | 93.25 | **94.42** | 93.53 | 94.13 |
| | | training param. (M) | 1.73 | N/A | 3.47 | 3.47 | 5.21 | 5.21 | 10.39 | 10.39 |
| | | testing param. (M) | 1.73 | 1.73 | 1.73 | 1.73 | 1.73 | 1.73 | 1.73 | 1.73 |
| CIFAR 100 | 32 | test accuracy (%) | 68.17 | N/A | 68.84 | 70.32 | 67.84 | 70.25 | 69.23 | **70.53** |
| | | training param. (M) | 0.47 | N/A | 0.95 | 0.95 | 1.44 | 1.44 | 2.84 | 2.84 |
| | | testing param. (M) | 0.46 | 0.46 | 0.46 | 0.46 | 0.46 | 0.46 | 0.46 | 0.46 |
| | 56 | test accuracy (%) | 70.25 | 70.78 | 70.6 | 72.79 | 69.62 | 72.28 | 70.69 | **73.01** |
| | | training param. (M) | 0.86 | N/A | 1.73 | 1.73 | 2.60 | 2.60 | 5.17 | 5.17 |
| | | testing param. (M) | 0.86 | 0.86 | 0.86 | 0.86 | 0.86 | 0.86 | 0.86 | 0.86 |
| | 110 | test accuracy (%) | 72.11 | 72.22 | 72.33 | 73.98 | 71.28 | **74.17** | 71.7 | 73.69 |
| | | training param. (M) | 1.73 | N/A | 3.48 | 3.48 | 5.22 | 5.22 | 10.40 | 10.40 |
| | | testing param. (M) | 1.73 | 1.73 | 1.73 | 1.73 | 1.73 | 1.73 | 1.73 | 1.73 |

\* Validation accuracies from Cao et al. (2020) are averages over the last five epochs across five training runs.

## 5 OVERCOMPLETE KNOWLEDGE DISTILLATION

In contrast to compressed model training, in *knowledge distillation* (KD) we have the capacity to train a large model but want to deploy a small one. We study what we call *overcomplete* KD, in which a network is over-parameterized in such a way that an equivalent small model can be directly recovered. Similar approaches have been previously studied only with small models (Arora et al., 2018; Guo et al., 2020) or using convolution-specific methods (Ding et al., 2019; Cao et al., 2020). We take a simple factorization approach in which we decompose weights $W \in \mathbb{R}^{m \times n}$ to be products of 2 or 3 matrices while *increasing* the parameter count, either via depth or width. Here we consider three cases: the *full* (rank) setting where $W = UV^T$ for $U \in \mathbb{R}^{m \times m}, V \in \mathbb{R}^{n \times m}$, the *deep* setting where $W = UMV^T$ for $U, M \in \mathbb{R}^{m \times m}, V \in \mathbb{R}^{n \times m}$, and the *wide* setting where $W = UV^T$ for $U \in \mathbb{R}^{m \times 3m}, V \in \mathbb{R}^{n \times 3m}$. As before, we factorize all but the first and last layer and train factorized networks using the same routine as for the base model, except when replacing weight-decay by FD. We do not study SI here, using the default initialization for $U, V^T$ and setting $M = I_m$.

Table 3 shows the strength of this approach: we can train ResNet32 to beat ResNet56 and ResNet56 to beat ResNet110 while needing about the same number of parameters during training and only half as many at inference. Note that training ResNet56 using the "full" setting is also 1.5x faster than training ResNet110 (see Table 8 for timings). Furthermore, we improve substantially upon DO-Conv (Cao et al., 2020), which obtains much smaller improvements over the unfactorized baseline. In fact, Table 9 suggests our approach compares favorably even with regular KD methods; combined with its simplicity and efficiency this suggests that our overcomplete method can be considered as a baseline in the field. Finally, we also show that Frobenius decay is critical: without it, overcomplete KD performs worse than regular training. A detailed visualization of this, comparing the CIFAR performance of factorized ResNets across a range of rank-scale settings covering both the current high-rank (distillation) case and the previous low-rank (compression) case, can be found in Figure 2.

## 6 MULTI-HEAD ATTENTION AS FACTORIZED QUADRATIC FORMS

Our final setting is multi-head attention (Transformer) architectures (Vaswani et al., 2017). As discussed in Section 2, the MHA component is already a factorized layer, consisting of an aggregation over the output of two quadratic forms per head: the "Query-Key" (QK) forms passed into the softmax and the "Output-Value" (OV) forms that multiply the resulting attention scores (c.f. Equation 1). Transformers also contain large linear layers, which can also be factorized for efficiency.

**Improving Transformer Training and Compression:** We start with the original Transformer architecture on the IWSLT-14 translation task (Cettolo et al., 2014) but use an SGD-based training routine as a baseline (Gehring et al., 2017). As there is no weight-decay by default, we first tune both this and FD on the non-factorized model; here FD is applied to the *implicitly* factorized MHA layers.

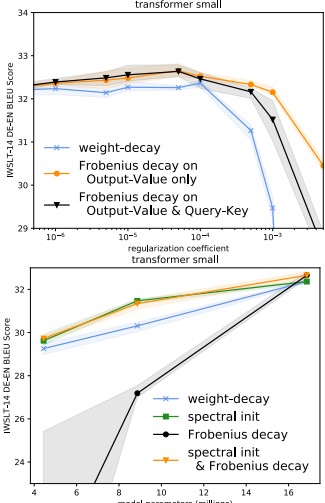

Figure 4: Transformer performance on IWSLT-14 as a function of regularization (top) and compression (bottom).

Table 4: Comparison of BERT unsupervised pretraining using LAMB (You et al., 2020) and our Frobenius decay scheme FLAMBé. Evaluation is conducted by fine-tuning the final model on the SQuAD question-answering task (Rajpurkar et al., 2016). Guided by IWSLT results, FLAMBé is applied only to the Output-Value form in MHA; regular weight-decay is used on all other parameters. Spectral initialization of MHA is used for FLAMBé as well, but we find the effect minimal. Regularization coefficients for both methods were obtained by tuning on the uncompressed model, targeting the unsupervised loss.

| model | optimizer | compression rate (MHA-only) | param. count (M) | SQuAD F1 | EM |
|---|---|---|---|---|---|
| BERT base | LAMB | 1.0 | 110 | 88.55 | 81.06 |
| | LAMB (SI & FD) | 1.0 | 110 | 88.16 | 80.86 |
| | FLAMBé (SI & FD) | 1.0 | 110 | 88.88 | 81.54 |
| | LAMB | 0.5 | 95.7 | 87.35 | 79.83 |
| | FLAMBé (SI & FD) | 0.5 | 95.7 | 87.69 | 80.42 |
| BERT large | LAMB | 1.0 | 340 | 91.41 | 84.99 |
| | FLAMBé (SI & FD) | 0.5 | 295.8 | 90.55 | 83.82 |

In Figure 4 we show that this alone yields an improvement: whereas the effect of weight-decay is either negligible or negative, tuning FD does improve the BLEU score. Furthermore, we see that tuning just the OV form in MHA is more robust at higher regularization levels than tuning both OV and QK. We conclude by examining both SI and FD when reducing the number of parameters by (1) factorizing all linear and embedding layers and (2) scaling down the embedding dimension in MHA. In Figure 4 we see that the benefit of Frobenius decay disappears when compressing; on the other hand, SI provides a strong boost under both types of decay, and is in fact necessary for FD to work at all. Note that the major effect here is for the factorized linear layers—we found that SI has minimal effect when applied to MHA, likely because those initializations have already been tuned.

**FLAMBé for Unsupervised BERT Pre-Training:** Lastly, we examine BERT (Devlin et al., 2019), a large transformer trained on a massive unsupervised text corpus and evaluated on downstream language tasks. The state-of-the-art training approach is via the LAMB optimizer (You et al., 2020) using weight-decay based on the AdamW algorithm of Loshchilov & Hutter (2019), in which $\lambda\eta$ times each parameter is subtracted from itself; this is equivalent to $\ell_2$-regularization for SGD but not for adaptive methods. We can define a similar Frobenius alternative by subtracting $\lambda\eta$ times the Frobenius gradients $\boldsymbol{U}\boldsymbol{V}^T\boldsymbol{V} = \nabla_{\boldsymbol{U}}\frac{1}{2}\|\boldsymbol{U}\boldsymbol{V}^T\|_F^2$ and $\boldsymbol{V}\boldsymbol{U}^T\boldsymbol{U} = \nabla_{\boldsymbol{V}}\frac{1}{2}\|\boldsymbol{U}\boldsymbol{V}^T\|_F^2$ from $\boldsymbol{U}$ and $\boldsymbol{V}$, respectively; when used with the LAMB optimizer we call this method FLAMBé. We see in Table 4 that FLAMBé outperforms the simple FD modification of LAMB and, as with IWSLT, leads to an improvement in downstream task performance without changing model size. For BERT, however, applying FD via FLAMBé also leads to better downstream performance when scaling down the MHA embedding dimension by half. Besides achieving a better compressed model, the success of FLAMBé also shows the potential of new types of decay schemes for adaptive methods.

## 7 CONCLUSION

In this paper we studied the design of training algorithms for deep nets containing factorized layers, demonstrating that two simple specializations of standard initialization and regularization schemes from the unfactorized case lead to strong improvements for model compression, knowledge distillation, and MHA-based architectures. While we largely focused on the case where the unfactorized model uses Gaussian initialization and $\ell_2$-regularization, we believe our work provides guidance for the many cases where other schemes are used to enforce alternative priors or improve optimization. For example, SI as defined can be applied to any random initialization, while our FD results suggest that regularizers such as penalty terms and DropConnect (Wan et al., 2013) should be applied on the product matrix rather than directly on individual factors. The success of SI and FD for both low-rank and tensor decomposition also suggests that these schemes may be useful for other types of factorized neural layers, such as ACDC (Moczulski et al., 2015) or K-matrices (Dao et al., 2020).

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

## A  GENERALIZATION ERROR OF FACTORIZED LAYERS

In this section we briefly discuss how to apply the generalization bounds in Neyshabur et al. (2018) to factorized models.

**Definition A.1.** *For any $\gamma > 0$ and distribution $\mathcal{D}$ over classification data $\mathcal{X} \times \mathcal{Y}$ the $\gamma$-**margin-loss** of a model $f_\theta : \mathcal{X} \mapsto \mathcal{Y}$ is defined as $\ell_\mathcal{D}^\gamma(f_\theta) = \mathbb{P}_{(\boldsymbol{x}, y) \sim \mathcal{D}} (f_\theta(\boldsymbol{x})[y] \leqslant \gamma + \max_{y' \neq y} f_\theta(\boldsymbol{x})[y'])$.*

Note that $\ell_\mathcal{D}^0$ is the expected classification loss over the distribution and $\ell_{\text{Uniform}(\mathcal{S})}^\gamma$ is the empirical $\gamma$-margin-loss. We have the following corollaries:

**Corollary A.1.** *Let $f_\theta$ be a neural network with $L-1$ factorized fully-connected ReLU layers $g_i(U_i V_i^T, x_{i-1}) = \max\{U_i V_i^T x_{i-1}, 0_m\}$ of hidden dimension $m$ and one factorized classification layer $g_L(U_L V_L^T, x_{L-1}) = U_L V_L^T x_{L-1}$. Let $\mathcal{D}$ be a distribution over $B$-bounded classification data $\mathcal{X} \times \mathcal{Y}$ and suppose we have a finite set $\mathcal{S}$ of i.i.d. samples from it. If $\text{rank}(U_i V_i^T) \leqslant r$ and $\|U_i V_i^T\|_2\} \leqslant \sigma$ for all $i$ then for any $\delta > 0$ we have w.p. $1 - \delta$ that*

$$\ell_{\mathcal{D}}^0(f_\theta) \leqslant \ell_{\text{Uniform}(\mathcal{S})}^\gamma + \mathcal{O}\left(\sqrt{\frac{B^2 L^3 m \sigma^{2L} r \log(Lm) \prod_{i=1}^L \|U_i V_i^T\|_2^2 + \log\frac{L|\mathcal{S}|}{\delta}}{\gamma^2 |\mathcal{S}|}}\right) \tag{4}$$

*Proof.* Apply the inequality $\|W_i\|_F^2 / \|W_i\|_2^2 \leqslant \text{rank}(W_i)$ to Neyshabur et al. (2018, Theorem 1) and substitute $W_i = U_i V_i^T$. $\qquad\square$

**Corollary A.2.** *Let $f_\theta$ be a neural network with $L-1$ factorized fully-connected ReLU layers $g_i(U_i(\prod_{j=1}^d M_{ij})V_i^T, x_{i-1}) = \max\{U_i(\prod_{j=1}^d M_{ij})V_i^T x_{i-1}, 0_m\}$ of hidden dimension $m$ and one factorized classification layer $g_L(U_L(\prod_{j=1}^d M_{Lj})V_L^T, x_{L-1}) = U_L(\prod_{j=1}^d M_{Lj})V_L^T x_{L-1}$. Let $\mathcal{D}$ be a distribution over $B$-bounded classification data $\mathcal{X} \times \mathcal{Y}$ from which we have a finite set $\mathcal{S}$ of i.i.d. samples. If $\|U_i \prod_{j=1}^d M_{ij} V_i^T\|_2\} \leqslant \sigma$ for all $i$ then for any $\delta > 0$ we have w.p. $1 - \delta$ that*

$$\ell_{\mathcal{D}}^0(f_\theta) \leqslant \ell_{\text{Uniform}(\mathcal{S})}^\gamma + \mathcal{O}\left(\sqrt{\frac{B^2 L^2 m \sigma^{2L-2} \log(Lm) \sum_{i=1}^L \|U_i(\prod_{j=1}^d M_{ij})V_i^T\|_F^2 + \log\frac{L|\mathcal{S}|}{\delta}}{\gamma^2 |\mathcal{S}|}}\right) \tag{5}$$

*Proof.* Apply the equality $\left(\prod_{i=1}^L \|W_i\|_2^2\right) \sum_{i=1}^L \frac{\|W_i\|_F^2}{\|W_i\|_2^2} = \sum_{i=1}^L \|W_i\|_F^2 \prod_{j\neq i} \|W_j\|_2^2$ to Neyshabur et al. (2018, Theorem 1) and substitute $W_i = U_i(\prod_{j=1}^d M_{ij})V_i^T$. $\qquad\square$

# B  PROOF OF CLAIM 3.1

*Proof.* Let $\rho_t = \|W_t\|_F$ be the Frobenius norm of the composed matrix at time $t$. Applying the update rules for $U_t$ and $V_t$ and using the fact that $\rho_t \nabla_{W_t} = \nabla_{\hat{W}_t}$ yields

$$\begin{aligned}
W_{t+1} = (U_t - \eta \nabla_{U_t})(V_t^T - \eta \nabla_{V_t^T}) &= (U_t - \eta \nabla_{W_t} V_t)(V_t^T - \eta U_t^T \nabla_{W_t^T}) \\
&= W_t - \frac{\eta}{\rho_t}\hat{\nabla}_t + \eta^2 \nabla_{W_t} W_t^T \nabla_{W_t}
\end{aligned} \tag{6}$$

Taking the squared norm of both sides yields $\rho_{t+1}^2 = \rho_t^2 - 2\eta \text{Tr}(\hat{W}_t^T \hat{\nabla}_t) + \mathcal{O}(\eta^2)$; we can then take the square root of both sides and use a Taylor expansion to obtain

$$\rho_{t+1} = \rho_t \sqrt{1 - \frac{2\eta}{\rho_t^2}\text{Tr}(\hat{W}_t^T \hat{\nabla}_t) + \mathcal{O}(\eta^2)} = \rho_t - \frac{\eta}{\rho_t}\text{Tr}(\hat{W}_t^T \hat{\nabla}_t) + \mathcal{O}(\eta^2) \tag{7}$$

Then, starting from Equation 6 divided by $\rho_{t+1}$, substituting Equation 7, and applying a Taylor expansion yields

$$\begin{aligned}
\hat{W}_{t+1} &= \frac{\rho_t}{\rho_{t+1}}\hat{W}_t - \frac{\eta}{\rho_t \rho_{t+1}}\hat{\nabla}_t + \mathcal{O}(\eta^2) \\
&= \frac{1}{1 - \frac{\eta}{\rho_t^2}\text{Tr}(\hat{W}_t^T \hat{\nabla}_t)}\hat{W}_t - \frac{\eta}{\rho_t^2 - \eta \text{Tr}(\hat{W}_t^T \hat{\nabla}_t)}\hat{\nabla}_t + \mathcal{O}(\eta^2) \\
&= \left(1 + \frac{\eta}{\rho_t^2}\text{Tr}(\hat{W}_t^T \hat{\nabla}_t)\right)\hat{W}_t - \frac{\eta}{\rho_t^2}\hat{\nabla}_t + \mathcal{O}(\eta^2) \\
&= \hat{W}_t - \frac{\eta}{\rho_t^2}(\hat{\nabla}_t - (\hat{w}_t^T \text{vec}(\hat{\nabla}_t))\hat{W}_t) + \mathcal{O}(\eta^2)
\end{aligned} \tag{8}$$

Vectorizing and observing that $(\hat{w}_t^T \text{vec}(\hat{\nabla}_t))\hat{w}_t = \hat{w}_t \hat{w}_t^T \text{vec}(\hat{\nabla}_t)$ yields the result. $\qquad\square$

## C EXPERIMENTAL DETAILS FOR TRAINING CONVOLUTIONAL NETWORKS

For experiments with regular ResNets on CIFAR we used code provided here: `https://github.com/akamaster/pytorch_resnet_cifar10`. All hyperparameter settings are the same, except initialization and regularization as appropriate, with the exception that we use a warmup epoch with a 10 times smaller learning rate for ResNet56 for stability (this is already done by default for ResNet110).

For comparisons with sparse model training of ResNet32$^{x2}$ and VGG19$^{NFC}$ we use code by Wang et al. (2019, `https://github.com/alecwangcq/EigenDamage-Pytorch`), which is closely related to that of the lottery-ticket-guessing paper by Wang et al. (2020). All hyperparameter settings are the same, except (1) initialization and regularization as appropriate and (2) for Tiny-ImageNet we only train for 200 epochs instead of 300.

For comparisons with tensor decomposition training of WideResNet we use code by Gray et al. (2019, `https://github.com/BayesWatch/deficient-efficient`). All hyperparameter settings are the same, except initialization and regularization as appropriate.

## D EXPERIMENTAL DETAILS FOR TRAINING TRANSFORMER MODELS

For experiments with Transformer models on machine translation we used code provided here: `https://github.com/StillKeepTry/Transformer-PyTorch`. All hyperparameter settings are the same, except initialization and regularization as appropriate.

For comparisons with LAMB optimization of BERT, we use an implementation provided by NVIDIA: `https://github.com/NVIDIA/DeepLearningExamples/tree/master/PyTorch/LanguageModeling/BERT`. All hyperparameter settings are the same, except initialization and regularization as appropriate. For fine-tuning on SQuAD we apply the same optimization routine to all pre-trained models.

## E PAST WORK ON KNOWLEDGE DISTILLATION

We first briefly summarize past work on overcomplete KD. While the use of factorized neural layers for improving training has theoretical roots (Arora et al., 2018; Du & Hu, 2019), we are aware of two other works focusing on experimental practicality: ExpandNets (Guo et al., 2020) and DO-Conv (Cao et al., 2020). As the former focuses on small student networks, numerically we compare directly to the latter, showing much better improvement due to distillation for both ResNet56 and ResNet110 on both CIFAR-10 and CIFAR-100 in Table 3. Note that we are also aware of one paper proposing a related method that also trains an over-parameterized model without increasing expressivity that can then be collapsed into a smaller "original" model (Ding et al., 2019); their approach, called ACNet, passes the input to each layer through differently-shaped kernels that can be composed additively. Note that it is unclear how to express this method as a factorization and it may not be easy to generalize to non-convolutional networks, so we do not view it as an overcomplete KD approach.

We now conduct a brief comparison with both these works and more standard approaches to KD. Direct comparison with past work is made difficult by the wide variety of training routines, teacher models, student models, and evaluation procedures employed by the community. Comparing our specific approach is made even more difficult by the fact that we have no teacher network, or at least not one that is a standard model used in computer vision. Nevertheless, in Table 9 we collect an array of existing results that can be plausibly compared to our own overcomplete distillation of ResNets. Even here, note that absolute numbers vary significantly, so we focus on changes in accuracy. As can be seen from the results, our overcomplete approach yields the largest improvements for ResNet56 and ResNet110 on both CIFAR-10 and CIFAR-100. For ResNet32, the Snapshot Distillation method of Xu & Liu (2019) outperforms our own, although it does not do so for ResNet110 and is not evaluated for ResNet56. On CIFAR-10 the additive ACNet approach also has a larger performance improvement for ResNet32. Nevertheless, our method is still fairly close in these cases, so the results in Table 9 together with the simplicity and short (single-stage) distillation routine of our overcomplete approach suggest that it should be a standard baseline for KD.

Table 5: Pruning, sparse training, and low-rank training for VGG-19$^{\text{NFC}}$, i.e. the model of Simonyan & Zisserman (2015) with no fully-connected layers, as in Wang et al. (2020).

| **CIFAR-10** | uncompressed | 0.1 | 0.05 | 0.02 |
|---|---|---|---|---|
| baseline* | 94.23 | | | |
| baseline[†] | 93.70 | | | |
| baseline (reproduced) | 93.75±0.48 | | | |
| OBD* LeCun et al. (1990) | | 93.74 | 93.58 | 93.49 |
| MLPrune* (Zeng & Urtasun, 2019) | | 93.83 | 93.69 | 93.49 |
| LT*, original initialization | | 93.51 | 92.92 | 92.34 |
| LT*, reset to epoch 5 | | 93.82 | 93.61 | 93.09 |
| LR Rewinding[†] (Renda et al., 2020) | | 94.14±0.17 | 93.99±0.15 | 10.00±0.00 |
| Hybrid Tickets[†] (Su et al., 2020) | | 94.00±0.12 | 93.83±0.10 | 93.52±0.28 |
| DSR* (Mostafa & Wang, 2019) | | 93.75 | 93.86 | 93.13 |
| SET* (Mocanu et al., 2018) | | 92.46 | 91.73 | 89.18 |
| Deep-R* (Bellec et al., 2018) | | 90.81 | 89.59 | 86.77 |
| SNIP* (Lee et al., 2018) | | 93.63±0.06 | 93.43±0.20 | 92.05±0.28 |
| GraSP* (Wang et al., 2020) | | 92.59±0.10 | 91.01±0.21 | 87.51±0.31 |
| Random Tickets[†] (Su et al., 2020) | | 93.77±0.10 | 93.42±0.22 | 92.45±0.22 |
| Low-Rank | | 90.71±0.42 | 88.08±0.14 | 80.81±0.85 |
| Low-Rank (SI) | | 90.99±0.34 | 88.61±0.95 | 82.53±0.14 |
| Low-Rank (FD) | | 91.57±0.26 | 88.16±0.50 | 82.81±0.37 |
| Low-Rank (SI & FD) | | 91.58±0.27 | 88.98±0.18 | 83.27±0.56 |
| **CIFAR-100** | uncompressed | 0.1 | 0.05 | 0.02 |
| baseline* | 74.16 | | | |
| baseline[†] | 72.60 | | | |
| baseline (reproduced) | 73.42±0.17 | | | |
| OBD* LeCun et al. (1990) | | 73.83 | 71.98 | 67.79 |
| MLPrune* (Zeng & Urtasun, 2019) | | 73.79 | 73.07 | 71.69 |
| LT*, original initialization | | 72.78 | 71.44 | 68.95 |
| LT*, reset to epoch 5 | | 74.06 | 72.87 | 70.55 |
| LR Rewinding[†] (Renda et al., 2020) | | 73.73±0.18 | 72.39±0.40 | 1.00±0.00 |
| Hybrid Tickets[†] (Su et al., 2020) | | 73.53±0.20 | 73.10±0.11 | 71.61±0.46 |
| DSR* (Mostafa & Wang, 2019) | | 72.31 | 71.98 | 70.70 |
| SET* (Mocanu et al., 2018) | | 72.36 | 69.81 | 65.94 |
| Deep-R* (Bellec et al., 2018) | | 66.83 | 63.46 | 59.58 |
| SNIP* (Lee et al., 2018) | | 72.84±0.22 | 71.83±0.23 | 58.46±1.10 |
| GraSP* (Wang et al., 2020) | | 71.95±0.18 | 71.23±0.12 | 68.90±0.47 |
| Random Tickets[†] (Su et al., 2020) | | 72.55±0.14 | 71.37±0.09 | 68.98±0.34 |
| Low-Rank | | 62.32±0.26 | 54.97±5.28 | 42.80±3.14 |
| Low-Rank (SI) | | 63.77±0.93 | 58.51±1.90 | 45.02±3.24 |
| Low-Rank (FD) | | 64.07±1.58 | 59.34±2.97 | 45.73±2.62 |
| Low-Rank (SI & FD) | | 65.61±1.48 | 60.1±1.29 | 46.43±1.26 |
| **Tiny-ImageNet** | uncompressed | 0.1 | 0.05 | 0.02 |
| baseline* | 61.38 | | | |
| baseline[†] | 61.57 | | | |
| baseline (reproduced) | 62.19±0.74 | | | |
| OBD* LeCun et al. (1990) | | 61.21 | 60.49 | 54.98 |
| MLPrune* (Zeng & Urtasun, 2019) | | 60.23 | 59.23 | 55.55 |
| LT*, original initialization | | 60.32 | 59.48 | 55.12 |
| LT*, reset to epoch 5 | | 61.19 | 60.57 | 56.18 |
| DSR* (Mostafa & Wang, 2019) | | 62.43 | 59.81 | 58.36 |
| SET* (Mocanu et al., 2018) | | 62.49 | 59.42 | 56.22 |
| Deep-R* (Bellec et al., 2018) | | 55.64 | 52.93 | 49.32 |
| SNIP* (Lee et al., 2018) | | 61.02±0.41 | 59.27±0.39 | 48.95±1.73 |
| GraSP* (Wang et al., 2020) | | 60.76±0.23 | 59.50±0.33 | 57.28±0.34 |
| Random Tickets[†] (Su et al., 2020) | | 60.94±0.38 | 59.48±0.43 | 55.87±0.16 |
| Low-Rank | | 48.50±3.43 | 46.98±1.68 | 36.57±1.03 |
| Low-Rank (SI) | | 49.87±1.39 | 46.73±2.83 | 37.25±0.19 |
| Low-Rank (FD) | | 53.93±0.66 | 47.95±1.13 | 35.63±0.71 |
| Low-Rank (SI & FD) | | 53.53±0.45 | 47.60±0.51 | 35.68±0.64 |

\* Obtained from Wang et al. (2020).
[†] Obtained from Su et al. (2020).

Table 6: Pruning, sparse training, and low-rank training for ResNet-32$^{x2}$, i.e. the model of He et al. (2016) with twice the number of filters, as in Wang et al. (2020).

| CIFAR-10 | uncompressed | 0.1 | 0.05 | 0.02 |
|---|---|---|---|---|
| baseline* | 94.80 | | | |
| baseline[†] | 94.62 | | | |
| baseline (reproduced) | 94.49±0.29 | | | |
| OBD* LeCun et al. (1990) | | 94.17 | 93.29 | 90.31 |
| MLPrune* (Zeng & Urtasun, 2019) | | 94.21 | 93.02 | 89.65 |
| LT*, original initialization | | 92.31 | 91.06 | 88.78 |
| LT*, reset to epoch 5 | | 93.97 | 92.46 | 89.18 |
| LR Rewinding[†] (Renda et al., 2020) | | 94.14±0.10 | 93.02±0.28 | 90.83±0.22 |
| Hybrid Tickets[†] (Su et al., 2020) | | 93.98±0.15 | 92.96±0.13 | 90.85±0.06 |
| DSR* (Mostafa & Wang, 2019) | | 92.97 | 91.61 | 88.46 |
| SET* (Mocanu et al., 2018) | | 92.30 | 90.76 | 88.29 |
| Deep-R* (Bellec et al., 2018) | | 91.62 | 89.84 | 86.45 |
| SNIP* (Lee et al., 2018) | | 92.59±0.10 | 91.01±0.21 | 87.51±0.31 |
| GraSP* (Wang et al., 2020) | | 92.38±0.21 | 91.39±0.25 | 88.81±0.14 |
| Random Tickets[†] (Su et al., 2020) | | 92.97±0.05 | 91.60±0.26 | 89.10±0.33 |
| Low-Rank | | 93.59±0.12 | 92.45±0.69 | 87.95±0.91 |
| Low-Rank (SI) | | 92.52±0.66 | 91.72±0.63 | 87.36±0.65 |
| Low-Rank (FD) | | 92.92±0.29 | 89.44±0.25 | 83.05±1.02 |
| Low-Rank (SI & FD) | | 94.34±0.34 | 92.90±0.43 | 87.97±0.86 |
| **CIFAR-100** | uncompressed | 0.1 | 0.05 | 0.02 |
| baseline* | 74.64 | | | |
| baseline[†] | 74.57 | | | |
| baseline (reproduced) | 75.41±1.26 | | | |
| OBD* LeCun et al. (1990) | | 71.96 | 68.73 | 60.65 |
| MLPrune* (Zeng & Urtasun, 2019) | | 72.34 | 67.58 | 59.02 |
| LT*, original initialization | | 68.99 | 65.02 | 57.37 |
| LT*, reset to epoch 5 | | 71.43 | 67.28 | 58.95 |
| LR Rewinding[†] (Renda et al., 2020) | | 72.41±0.49 | 67.22±3.42 | 59.22±1.15 |
| Hybrid Tickets[†] (Su et al., 2020) | | 71.47±0.26 | 69.28±0.40 | 63.44±0.34 |
| DSR* (Mostafa & Wang, 2019) | | 69.63 | 68.20 | 61.24 |
| SET* (Mocanu et al., 2018) | | 69.66 | 67.41 | 62.25 |
| Deep-R* (Bellec et al., 2018) | | 66.78 | 63.90 | 58.47 |
| SNIP* (Lee et al., 2018) | | 68.89±0.45 | 65.22±0.69 | 54.81±1.43 |
| GraSP* (Wang et al., 2020) | | 69.24±0.24 | 66.50±0.11 | 58.43±0.43 |
| Random Tickets[†] (Su et al., 2020) | | 69.70±0.48 | 66.82±0.12 | 60.11±0.16 |
| Low-Rank | | 72.71±0.43 | 67.86±0.91 | 61.08±0.76 |
| Low-Rank (SI) | | 71.71±1.46 | 67.41±1.12 | 60.79±0.26 |
| Low-Rank (FD) | | 71.85±0.39 | 66.89±0.73 | 55.31±0.61 |
| Low-Rank (SI & FD) | | 74.41±0.67 | 70.22±0.64 | 61.40±0.96 |
| **Tiny-ImageNet** | uncompressed | 0.15 | 0.1 | 0.1 |
| baseline* | 62.89 | | | |
| baseline[†] | 62.92 | | | |
| baseline (reproduced) | 63.02±0.94 | | | |
| OBD* LeCun et al. (1990) | | 58.55 | 56.80 | 51.00 |
| MLPrune* (Zeng & Urtasun, 2019) | | 58.86 | 57.62 | 51.70 |
| LT*, original initialization | | 56.52 | 54.27 | 49.47 |
| LT*, reset to epoch 5 | | 60.31 | 57.77 | 51.21 |
| DSR* (Mostafa & Wang, 2019) | | 57.08 | 57.19 | 56.08 |
| SET* (Mocanu et al., 2018) | | 57.02 | 56.92 | 56.18 |
| Deep-R* (Bellec et al., 2018) | | 53.29 | 52.62 | 52.00 |
| SNIP* (Lee et al., 2018) | | 56.33±0.24 | 55.43±0.14 | 49.57±0.44 |
| GraSP* (Wang et al., 2020) | | 57.25±0.11 | 55.53±0.11 | 51.34±0.29 |
| Random Tickets[†] (Su et al., 2020) | | N/A | 55.26±0.22 | 51.41±0.38 |
| Low-Rank | | 60.82±0.72 | 58.72±0.53 | 55.39±1.02 |
| Low-Rank (SI) | | 59.53±1.41 | 57.60±0.88 | 55.00±0.90 |
| Low-Rank (FD) | | 59.45±0.82 | 57.24±0.61 | 54.15±1.22 |
| Low-Rank (SI & FD) | | 62.24±0.39 | 60.25±1.02 | 55.97±0.48 |

\* Obtained from Wang et al. (2020).
[†] Obtained from Su et al. (2020).

Table 7: Comparison of low-rank and tensor-decomposition training (mean of 3 trials) of WideResNet28-10 on CIFAR-10. The best error for each compression-decomposition setting is **bolded**; in addition, the best error at each compression level is underlined.

| compression | decomposition | decay ($\lambda = 0.005$) | initialization default | spectral |
|---|---|---|---|---|
| none ($\approx$36.5 param.) | | | **3.21$\pm$0.22** | |
| 0.01667 ($\approx$0.61M param.) | Low-Rank | regular | 7.62$\pm$0.08 | 7.72$\pm$0.11 |
| | | CRS* | 8.63$\pm$0.87 | 8.83$\pm$0.17 |
| | | Frobenius | **7.23$\pm$0.32** | 7.26$\pm$0.45 |
| | Tensor-Train | regular | 8.55$\pm$4.22 | 7.72$\pm$0.46 |
| | | CRS* | 7.01$\pm$0.35 | 6.52$\pm$0.17 |
| | | Frobenius | 7.26$\pm$0.73 | **6.22$\pm$0.50** |
| | Tucker | regular | 10.67$\pm$7.68 | 9.91$\pm$6.71 |
| | | CRS* | 8.28$\pm$0.05 | **6.79$\pm$0.34** |
| | | Frobenius | 8.65$\pm$0.27 | 7.07$\pm$0.28 |
| 0.06667 ($\approx$2.4M param.) | Low-Rank | regular | 5.31$\pm$0.06 | 5.49$\pm$0.42 |
| | | CRS* | 5.86$\pm$0.22 | 5.53$\pm$0.72 |
| | | Frobenius | 3.89$\pm$0.17 | **3.86$\pm$0.22** |
| | Tensor-Train | regular | 7.33$\pm$0.27 | 6.49$\pm$0.30 |
| | | CRS* | 5.20$\pm$0.37 | **4.79$\pm$0.15** |
| | | Frobenius | 5.02$\pm$0.15 | 5.10$\pm$0.35 |
| | Tucker | regular | 5.64$\pm$0.19 | 5.61$\pm$0.46 |
| | | CRS* | 5.30$\pm$0.35 | **5.15$\pm$0.23** |
| | | Frobenius | 5.30$\pm$0.12 | 5.24$\pm$0.21 |

[*] Regular weight-decay with coefficient scaled by the compression rate (Gray et al., 2019).

Table 8: Overcomplete ResNet performance (mean of 3 trials). The best accuracy at each depth is **bolded**; cases where we match the next deeper network with around the same *training* memory are underlined. Note that training times are reported for roughly similar machine types; all ResNet56 and ResNet110 times are reported on identical machine types.

| **CIFAR-10** | | | factorization type | | |
|---|---|---|---|---|---|
| depth | decay type/metric | unfactorized | full | deep | wide |
| | WD test accuracy | $92.66 \pm 0.34$ | $92.11 \pm 0.36$ | $91.37 \pm 0.09$ | $92.26 \pm 0.35$ |
| | training time (H) | $1.51 \pm 0.17$ | $1.86 \pm 0.23$ | $2.10 \pm 0.14$ | $2.20 \pm 0.22$ |
| 32 | FD test accuracy | | $93.32 \pm 0.07$ | $93.11 \pm 0.13$ | $\mathbf{93.36 \pm 0.11}$ |
| | training time (H) | | $1.60 \pm 0.22$ | $2.40 \pm 0.16$ | $1.97 \pm 0.23$ |
| | training param. (M) | 0.46 | 0.95 | 1.43 | 2.84 |
| | WD test accuracy | $93.15 \pm 0.56$ | $92.95 \pm 0.24$ | $92.52 \pm 0.13$ | $93.08 \pm 0.23$ |
| | training time (H) | $1.39 \pm 0.04$ | $2.12 \pm 0.00$ | $2.45 \pm 0.12$ | $3.51 \pm 0.09$ |
| 56 | FD test accuracy | | $\mathbf{\underline{94.13 \pm 0.16}}$ | $93.89 \pm 0.13$ | $93.83 \pm 0.10$ |
| | training time (H) | | $2.22 \pm 0.07$ | $2.89 \pm 0.05$ | $3.65 \pm 0.14$ |
| | training param. (M) | 0.85 | 1.72 | 2.59 | 5.16 |
| | WD test accuracy | $93.61 \pm 0.28$ | $93.73 \pm 0.16$ | $93.25 \pm 0.07$ | $93.53 \pm 0.07$ |
| | training time (H) | $3.51 \pm 0.06$ | $3.86 \pm 0.13$ | $4.92 \pm 0.15$ | $6.37 \pm 0.21$ |
| 110 | FD test accuracy | | $94.28 \pm 0.05$ | $\mathbf{94.42 \pm 0.09}$ | $94.13 \pm 0.05$ |
| | training time (H) | | $4.18 \pm 0.04$ | $5.77 \pm 0.22$ | $6.55 \pm 0.18$ |
| | training param. (M) | 1.73 | 3.47 | 5.21 | 10.39 |
| **CIFAR-100** | | | factorization type | | |
| depth | metric | unfactorized | full | deep | wide |
| | WD test accuracy | $68.17 \pm 0.48$ | $68.84 \pm 0.27$ | $67.84 \pm 0.50$ | $69.23 \pm 0.32$ |
| | training time (H) | $1.60 \pm 0.21$ | $1.80 \pm 0.23$ | $1.61 \pm 0.04$ | $2.48 \pm 0.19$ |
| 32 | FD test accuracy | | $70.32 \pm 0.41$ | $70.25 \pm 0.46$ | $\mathbf{70.53 \pm 0.07}$ |
| | training time (H) | | $1.61 \pm 0.10$ | $2.13 \pm 0.20$ | $2.61 \pm 0.18$ |
| | training param. (M) | 0.47 | 0.95 | 1.44 | 2.84 |
| | WD test accuracy | $70.25 \pm 0.30$ | $70.6 \pm 0.53$ | $69.62 \pm 0.35$ | $70.69 \pm 0.21$ |
| | training time (H) | $1.39 \pm 0.04$ | $2.09 \pm 0.04$ | $2.59 \pm 0.04$ | $3.48 \pm 0.27$ |
| 56 | FD test accuracy | | $72.79 \pm 0.37$ | $72.28 \pm 0.26$ | $\mathbf{73.01 \pm 0.39}$ |
| | training time (H) | | $2.30 \pm 0.05$ | $2.96 \pm 0.17$ | $3.33 \pm 0.19$ |
| | training param. (M) | 0.86 | 1.73 | 2.60 | 5.17 |
| | WD test accuracy | $72.11 \pm 0.37$ | $72.33 \pm 0.57$ | $71.28 \pm 0.62$ | $71.70 \pm 0.55$ |
| | training time (H) | $3.87 \pm 0.08$ | $3.78 \pm 0.04$ | $4.95 \pm 0.23$ | $6.12 \pm 0.25$ |
| 110 | FD test accuracy | | $73.98 \pm 0.22$ | $\mathbf{74.17 \pm 0.23}$ | $73.69 \pm 0.72$ |
| | training time (H) | | $4.46 \pm 0.07$ | $5.66 \pm 0.18$ | $6.81 \pm 0.18$ |
| | training param. (M) | 1.73 | 3.48 | 5.22 | 10.40 |

Table 9: Comparison of our knowledge distillation approach with past work in which the same student network attains roughly similar performance.

| CIFAR-10 baseline depth | method (source) | teacher | reported baseline accuracy | reported distilled accuracy | change in accuracy |
|---|---|---|---|---|---|
| 32 | SD (Xu & Liu, 2019) | self | 92.78 | 93.68 | **+0.90** |
| | ACNet (Ding et al., 2019) | additive | 94.31 | 95.09 | +0.78 |
| | wide factorization w. FD (ours) | overcomplete | 92.66 | 93.36 | +0.70 |
| 56 | KD-fn (Xu et al., 2020) | ResNet110 | 93.63 | 94.14 | +0.51 |
| | DO-Conv (Cao et al., 2020) | overcomplete | 93.26 | 93.38 | +0.12 |
| | full factorization w. FD (ours) | overcomplete | 93.15 | 94.13 | **+0.98** |
| 110 | SD (Xu & Liu, 2019) | self | 94.00 | 94.43 | +0.43 |
| | DO-Conv (Cao et al., 2020) | overcomplete | 93.72 | 93.93 | +0.21 |
| | deep factorization w. FD (ours) | overcomplete | 93.61 | 94.42 | **+0.81** |
| **CIFAR-100** baseline depth | method (source) | teacher | reported baseline accuracy | reported distilled accuracy | change in accuracy |
| 32 | CRD (Tian et al., 2020) | ResNet110 | 71.14 | 73.48 | +2.34 |
| | SD (Xu & Liu, 2019) | self | 68.99 | 71.78 | **+2.79** |
| | Snapshot (Yang et al., 2019) | self | 68.39 | 69.84 | +1.45 |
| | ACNet (Ding et al., 2019) | additive | 73.58 | 74.04 | +0.46 |
| | wide factorization w. FD (ours) | overcomplete | 68.17 | 70.53 | +2.36 |
| 56 | FT (Kim et al., 2018) | ResNet110 | 71.96 | 74.38 | +2.42 |
| | Snapshot (Yang et al., 2019) | self | 70.06 | 70.78 | +0.72 |
| | DO-Conv (Cao et al., 2020) | overcomplete | 70.15 | 70.78 | +0.63 |
| | wide factorization w. FD (ours) | overcomplete | 70.25 | 73.01 | **+2.76** |
| 110 | SD (Xu & Liu, 2019) | self | 73.21 | 74.96 | +1.75 |
| | Snapshot (Yang et al., 2019) | self | 71.47 | 72.48 | +1.01 |
| | DO-Conv (Cao et al., 2020) | overcomplete | 72.00 | 72.22 | +0.22 |
| | deep factorization w. FD (ours) | overcomplete | 72.11 | 74.17 | **+2.06** |

