# OpenReview forum: "Initialization and Regularization of Factorized Neural Layers"
_ICLR.cc/2021/Conference — ICLR 2021 Poster_

### Official Review · AnonReviewer1 · 2020-10-17
**Empirical result studying spectral initialization and Frobenius decay on factorized NN**

**Rating:** 6
**Confidence:** 3

**Review:**

This paper studies initialization and regularization in factorized neural networks (reparameterize a weight matrix by the product of several weight matrices). The authors proposed spectral initialization, that is to initialize the factorized matrices using the SVD of the un-factorized matrix. The authors also proposed Frobenius decay that is to regularize the Frobenius norm of the product of the factorized weight matrices. The motivation is to simulate the routines for non-decomposed counterparts. The authors empirically showed the effectiveness of spectral initialization and Frobenius decay in different applications: compressed model training, knowledge distillation, and multi-head self-training.

I think it’s important to study the initialization and regularization for factorized neural networks. A priori, it needs different initialization and regularization methods due to different architecture compared with its non-decomposed counter-part. This paper gave very simple and natural solutions and was able to show its effectiveness in experiments.

I also have some questions as below:
1. In the experiments in section 5 (knowledge distillation), default initialization is used instead of spectral initialization. I wonder if SI leads to a bad performance here. If that’s the case, it requires more explanation of why SI fails in this setting.
2. In Figure 1, it seems FD is a stronger regularizer compared with default weight decay. It seems if the regularization coefficient is carefully tuned for each regularizer, the benefits of FD is actually not very significant. Also, what’s "no decay (normalized)"?
3. In section 2, the definition of the factorized CNN is not very clear to me. It might be good to give more detailed definitions here.
4. Spectral initialization requires computing SVD of the weight matrix. If the matrix dimension is high, this step can be very time-consuming. I wonder if there is any more efficient way to construct the factorized matrices so that their product is still as i.i.d. Gaussian matrix. Because we don't need to compute the SVD for an arbitrary matrix, what we need is only to make sure that the product of the factorized matrixes is distributed as i.i.d. Gaussian.

---

> ### Author Response · Authors · 2020-11-16
> **Response to AnonReviewer1**
>
> Thank you for the positive review. We hope to address your questions below.
>
> 1. _“In the experiments in section 5 (knowledge distillation), default initialization is used instead of spectral initialization. I wonder if SI leads to a bad performance here. If that’s the case, it requires more explanation of why SI fails in this setting.”_
> We do not use SI in Section 5 because there is no obvious way to apply it to overcomplete factorizations (when the rank is higher than the original rank). For the full-rank case (the “full” setting), the median performance is roughly the same with and without SI. Here are the results for ResNet56:
> |       | CIFAR-10 | CIFAR-100 |
> |-------|----------|-----------|
> | WD    | 92.45    | 69.23     |
> | SI+WD | 92.49    | 70.42     |
> | FD    | 93.68    | 72.18     |
> | SI+FD | 93.48    | 72.19     |
>
> 2.
>    - _“In Figure 1, it seems FD is a stronger regularizer compared with default weight decay. It seems if the regularization coefficient is carefully tuned for each regularizer, the benefits of FD is actually not very significant.”_
> In Figure 1 [note: in the revision the relevant plot is now in Figure 2], tuning weight-decay from the default value (1E-4) shows little to no improvement for both ResNet20 and ResNet44, while tuning FD does yield a noticeable improvement on ResNet20. We thus view this plot as showing that we cannot achieve FD performance by tuning weight-decay, and that tuning weight-decay does not yield significant improvements. The gap between their performances is 0.5-1%, which is not too small at this accuracy on CIFAR-10, and the gaps in the other experiments are more significant (usually 1-2% in Table 1, 0.5-1.5% in Table 2 at very high accuracies, and usually 1-2% in Table 3). While we did not extensively tune weight-decay in those settings, we believe that (1) the lack of need of tuning FD is a significant advantage in a setting where we aim for efficient training and (2) the figure suggests tuning weight-decay does not help much anyway.
>    - _“[W]hat’s "no decay (normalized)"?”_
> “no decay (normalized)” refers to the setting where we first train a network using FD and save the Frobenius norm of the (recomposed) weight matrix UV^T at each layer after each SGD step; we then re-initialize and train the network with zero weight-decay, but after each SGD step we normalize U and V at each so that their product UV^T has the same Frobenius norm as that of the same layer at the same time step of the previous optimization. This experiment is described in the last paragraph of Section 3.3; in the revision we have added a note about this in the caption.
>
> 3. _“In section 2, the definition of the factorized CNN is not very clear to me. It might be good to give more detailed definitions here.”_
> We have added details about this in the revision (Section 2.2).
>
> 4. _“Spectral initialization requires computing SVD of the weight matrix. If the matrix dimension is high, this step can be very time-consuming. I wonder if there is any more efficient way to construct the factorized matrices so that their product is still as i.i.d. Gaussian matrix. Because we don't need to compute the SVD for an arbitrary matrix, what we need is only to make sure that the product of the factorized matrixes is distributed as i.i.d. Gaussian.”_
> Please see Point 2 of our general author response for a discussion of alternative initializations and the complexity of spectral initialization.

---

### Official Review · AnonReviewer4 · 2020-10-27
**An important research but few details are missing**

**Rating:** 6
**Confidence:** 3

**Review:**

This paper discusses about applying low-rank matrix and tensor factorization of weight and applying weight decay on them. This type of low-rank regularization is an important mechanism in deep learning models and already many researchers have shown interest in this topic, hence this paper would interest many researchers in the community.

The technical aspects of the paper seem to be correct. The experimental results are very encouraging since good improvements are shown with popular datasets. Also, the paper covers compression using ideas related to the state of the art tensor factorization methods.

The most important component in learning with factorization methods is specifying the appropriate rank of matrices and tensors. When using factorization methods problems such as matrix/tensor completion, the performance will strongly depend on the rank. It is not clear how the rank of weight matrices would affect the performance of the deep learning model. This paper does not address this issue. Furthermore, there are no theoretical results shown that take the rank into consideration. Can the authors add an experiment to show how different factorization with respect to ranks affects the performance? And/or can they give a theoretical result explaining how the rank relates to the improvement of the learning model?

---

> ### Author Response · Authors · 2020-11-16
> **Response to AnonReviewer4**
>
> Thank you for the positive review. We hope to address your questions below.
>
> - _“It is not clear how the rank of weight matrices would affect the performance of the deep learning model. This paper does not address this issue. […] Can the authors add an experiment to show how different factorization with respect to ranks affects the performance?”_
> Note that the experiments where we report performance as a function of compression rate or parameter count (e.g. Tables 1 and 2) effectively demonstrate the effect of rank on performance since the compression/parameter count is directly determined by varying the rank of the weight-matrices. However, in the revision we have added a plot (Figure 2, center and right) demonstrating the performance of ResNet as a function of the rank, showing that FD improves upon regular weight-decay consistently in both the low-rank and over-complete case (note that SI does not apply in the latter case).
>
> - _“[T]here are no theoretical results shown that take the rank into consideration. […]  [Can the authors] give a theoretical result explaining how the rank relates to the improvement of the learning model?”_
> In the low-rank case, standard learning theory suggests that reducing the size of the model class (e.g. by lowering the rank) can improve generalization error. There also exist modern generalization error bounds that improve with lower stable rank (and thus also rank since stable rank <= rank), c.f. Neyshabur et al.  (2018). Thus, since test error = training error + generalization error, the focus of our analysis is on improving the optimization of low-rank models in order the make the training error small, since these existing results already suggest that the generalization error will be small. We have slightly expanded upon this discussion in the revision (Section 2.1). For the high-rank (self-taught distillation) case, in the submission we pointed to recent work (Du & Hu, 2019) that shows that width can improve optimization of deep nets.

---

> > ### Comment · AnonReviewer4 · 2020-11-21
> > **Additional comments**
> >
> > Thank you for the reply! The new plot is helpful to understand the effect of the factored weight decay better.
> >
> > Regarding the second reply, it is understood that low rank regularization helps to improve performance. However, what I was indenting was if deep learning theory can be extended towards the proposed method. It would make the paper stronger if the existing generalization error bounds such as Neyshabur et al. (2018) can be extended in the paper.

---

> > > ### Author Response · Authors · 2020-11-22
> > > **Response to additional comments**
> > >
> > > Thank you for your message. The result of Neyshabur et al. (2018) can itself be extended to the factorized setting to justify the methods we propose. Specifically, they bound the 0-1 test loss of a $d$-layer fully connected ReLU net with weights $W\_1,\dots,W\_d$ by the empirical margin loss plus the following generalization error term with high probability (here $m$ is the number of samples):
> > >
> > > $$\tilde O\left(\sqrt{\frac{d^2\left(\prod\_{i=1}\^d\\|W\_i\\|\_2\^2\right)\sum\_{i=1}\^d\frac{\\|W\_i\\|\_F\^2}{\\|W\_i\\|\_2\^2}}m}\right)$$
> > >
> > > This bound can be used to directly obtain bounds for our setting by directly replacing all weights matrices $W\_i$ by their factorization $U\_iV\_i\^T$. For example, to account for the Frobenius decay penalty term $\frac12\\|U\_iV\_i\^T\\|\_F\^2$ that we propose as a regularizer, the above bound is equivalent to
> > >
> > > $$\tilde O\left(\sqrt{\frac{d^2\sum\_{i=1}\^d\\|U\_iV\_i\^T\\|\_F\^2\prod\_{j\ne i}\\|U\_jV\_j\^T\\|\_2\^2}m}\right)$$
> > >
> > > Thus controlling the magnitude of the squared Frobenius norm of each factorized layer can be interpreted as targeting a low generalization error. Alternatively, substituting the inequality $\\|W\\|\_F\^2/\\|W\\|\_2\^2\le rank(W)$ into the first bound and assuming all factorizations have rank $r$ yields a generalization error bound of
> > >
> > > $$\tilde O\left(\sqrt{\frac{d^3r\prod\_{i=1}\^d\\|U\_iV\_i\^T\\|\_2\^2}m}\right)$$
> > >
> > > We hope this latter point answers your original question about a theoretical explanation for how reducing the rank can improve the model learned.
> > >
> > > As with all theoretical results in deep learning, these bounds have limitations: they apply only to the margin loss, they do not account for batch-norm, and they are likely vacuous for practical sample sizes. However, we do not know of any work on generalization in deep learning that overcomes these limitations, and we believe such a result, if obtained, should be presented on its own as part of a paper on unfactorized deep nets, not as part of an analysis of the training routines of factorized models. For our purposes, we believe that these corollaries of Neyshabur et al. (2018) directly address your point that “[i]t would make the paper stronger if the existing generalization error bounds such as Neyshabur et al. (2018) can be extended in the paper.”
> > >
> > > In the most recent revision we have referenced these two bounds explicitly (using our paper’s notation and assuming for simplicity a bound on the spectral norm of each layer) in the main text (Sections 2.1 and 3.2) and stated them formally as corollaries of Theorem 1 of Neyshabur et al. (2018) in Appendix A.

---

> > > > ### Comment · AnonReviewer4 · 2020-11-25
> > > > **Follow-up comment**
> > > >
> > > > Thank you for the update and the explanation

---

### Official Review · AnonReviewer3 · 2020-10-27
**A good empirical study of known techniques for initilization and regularization**

**Rating:** 6
**Confidence:** 3

**Review:**

The paper studies how the initialization and regularization of the factorized layers $W=U\Pi_i M_i V^T$. It focused on the two known techniques, spectral initialization (SI) and Frobenius decay (FD) (i.e., regularize the weight $W$ rather than the factorized terms $U$, $V$, and $M_i$'s), to initialize and regularize such layers for training.
From the technical point of view, the paper is not novel, however, its strength is motivating the need for SI & FD, and providing various empirical studies in different applications such as model compression and self-taught distillation.
Strengths:
1. The paper is generally well-written and clear. The majority of claims are either cited appropriately or shown via extensive simulations.
2. Good motivations, supported by simulations (Figures 1, 2) and claim 3.1
3. Extensive simulations for different tasks: compressed model training, knowledge distillation, and multi-head attention, in different areas computer vision, question-answering, transformer training, and BERT.

Weaknesses:
The paper can be seen as merely applying known methods to different areas and observing its performance. It lefts many questions unanswered and raises some other concerns;
1. SI initialization is reasonable when training a compressed model from a "pre-trained" model since the decomposition error is minimized $W\approx UV^T$. However, when training from scratch, to my understanding, the method first generates a normal $W\sim N(0, \sigma^2 I)$, and uses SVD to find $U$ and $V$. SI + FD can be interpreted as assuming a Gaussian prior on $W$ and enforcing that prior on $W$ during training. One immediate question would be why such a prior is a good assumption in general? From VGG experiment (Table 1), one can conclude that specific initialization is not an important factor in training. Probably, for other simpler initializations for $U$ and $V$ (e.g., $U$ and random Rademacher and $V$ random normal, or even one of them being some truncated identity matrix) the FD will work too. The claims can be interpreted as 'If Gaussian prior is necessary for the parameters, SI+FD is the way for training."
2. Figure 1 shows the bound in equation (2) is tight for the compression. Is there any similar result for the distillation (i.e., $r>max(m,n)$)?
3. How to extend SI to have factorization $W=U(\Pi_{i=1}^d M_i)V^T$ for $d>0$?


There are a few minor points/typos that the paper can be improved on:
- The RHS equality in equation (2) is not trivial. It needs a proper reference or proof.
- In Table 1, Dynamic Sparsity for CIFAR100 at compression 0.02 performs better than SI&FD. The bold text in the table incorrectly implies that SI&FD is better.

---

> ### Author Response · Authors · 2020-11-16
> **Response to AnonReviewer3**
>
> Thank you for your positive review. We hope to address your questions below.
>
> - _“The paper can be seen as merely applying known methods to different areas and observing its performance.”_
> Please see Point 1 of our general author response for a discussion of novelty and contributions of the work.
>
> 1.
>    - _“SI + FD can be interpreted as assuming a Gaussian prior on W and enforcing that prior on W during training. One immediate question would be why such a prior is a good assumption in general? […] 'If Gaussian prior is necessary for the parameters, SI+FD is the way for training."”_
> Thank you for pointing this out. We agree that in general the Gaussian prior might not be a good assumption, and that its success in our experiments may be because it has been shown to be effective for training unfactorized models, thus making it reasonable to enforce it in the factorized case. However, if a different prior---implemented via some initialization distribution and some regularization term---is used in the unfactorized case, we believe our results suggest to (1) initialize using SVD on the full-rank matrix initialized with the same distribution and (2) applying the same regularization term on the recomposed matrix rather than the individual factors. So this would still be SI but the FD would be replaced by a different type of regularization. We have added a note about this in the revision (Section 7).
>    - _“From VGG experiment (Table 1), one can conclude that specific initialization is not an important factor in training.”_
> Even in the case of VGG, SI+FD improves upon FD alone on two of the three datasets. However, perhaps its more limited effect here does suggest that a different initialization could lead to further improvements.
>    - _“Probably, for other simpler initializations for U and V (e.g., U and random Rademacher and V random normal, or even one of them being some truncated identity matrix) the FD will work too.”_
> Yes, we do not have strong evidence that FD always requires SI to work well; in fact FD works well without it for the self-taught distillation experiments. Please also see Point 2 of our general author response for a discussion of alternative initializations.
>
> 2. _“Figure 1 shows the bound in equation (2) is tight for the compression. Is there any similar result for the distillation (i.e., r>max(m,n))?”_
> Thank you asking about this – we checked and found that the bound is indeed tight for the distillation case as well, suggesting that weight-decay is penalizing the nuclear norm across all ranks. We have included this result in the revision (Figure 1, right).
>
> 3. _“How to extend SI […] for d>0?.”_
> For deeper factorizations one can initialize the inner matrices M to be the identity (or a small perturbation). This keeps the distribution of the recomposed matrix the same. However, note that we only used d>0 in our self-taught distillation experiments, where we did not use SI since it is not clear how to apply it when the factorization is over-complete.
>
> - _“There are a few minor points/typos that the paper can be improved on.”_
> Thank you, both are fixed in the revision.

---

> > ### Comment · AnonReviewer3 · 2020-11-23
> > **additional comment**
> >
> > Thanks for the responses and update to the paper, esp. Fig. 1
> > Do you have any comment on why the gap between the upper bound and nuclear norm (equation 2) is high in the first few epochs and later becomes small? Is it because of the specific form of regularizer that automatically reduces the gap after few epochs of training?

---

> > > ### Author Response · Authors · 2020-11-24
> > > **Response to additional comment**
> > >
> > > Thank you for your message, and for your earlier suggestion to take a look at r>max{m,n}.
> > >
> > > Before addressing your question, note that there is not always a gap in the first few epochs, as it is initialization-dependent: at spectral initialization the factors U and V are orthogonal so the bound in Equation 2 starts out tight, while at i.i.d. Gaussian initialization the bound is loose. This is reflected in the Figure 1 plots (spectral is used in the middle, Gaussian on the left and right).
> > >
> > > Why the gap goes away after a few epochs of training is unclear, but it is unlikely to be due to the regularizer alone, because the bound is also tight when there is no regularization (the “no-decay” line in the left and center plots). One possible direction for explaining this phenomenon is via some implicit property of SGD, which could also account for the no-decay case.

---

### Official Review · AnonReviewer2 · 2020-10-29
**Technical contribution**

**Rating:** 6
**Confidence:** 3

**Review:**

This paper studies how to initialize via spectral initialization and regularize DNNs via Frobenius decay. The benefits of spectral initialization and Frobenius decay are demonstrated via many experiments. This paper focuses on the empirical study of both SI and FD.

My major concern is on its technical contribution. The spectral initialization is a scheme commonly used for low-rank models. It is well understood that spectral initialization is better than random initialization in low-rank literatures. Moreover, the proposed Frobenius decay is simply a penalization on the squared norm of the whole factorization. That is, it replaces the weight-decay $\|W\|_F^2$ in deep nets by plugging in the low-rank formulation of $W$. The behaviors of these two components in the experiments are expected. These two components have not provided much new insight.

~~~~~
This major concern is relieved after rebuttal.

---

> ### Author Response · Authors · 2020-11-16
> **Response to AnonReviewer2**
>
> Please see Point 1 in our general author response for the novelty and detailed contributions in our work. We would be happy to answer any questions about these claims or about the revision. Below we address some of your specific concerns about the two techniques we study.
>
> 1. _"It is well understood that spectral initialization is better than random initialization in low-rank literatures."_
> We would be interested in any references that compare spectral and random initialization so that we can cite them. The use of spectral initialization in past work is something we note in the second paragraph of Section 1, but to our knowledge these works do not make such comparisons and generally focus on initializing from pretrained full-rank networks rather than training from scratch.
>
> 2. _"Moreover, the proposed Frobenius decay is simply a penalization on the squared norm of the whole factorization.”_
> We believe simplicity is a benefit of the method. That such an effective and simple method is not more widely used makes it more necessary to formalize it and demonstrate its effectiveness.
>
> 3. _"The behaviors of these two components in the experiments are expected. These two components have not provided much new insight."_
> That the methods perform better is perhaps not surprising. However, (a) this behavior has not previously been analyzed mathematically and (b) we view many of our empirical findings as unexpected or at least not obvious, including the following:
>    - Weight-decay implicitly penalizes the nuclear norm of the product despite explicitly regularizing only a loose upper bound (Figure 1).
>    - Frobenius decay is better than weight-decay even after tuning the latter (Figures 2 and 3).
>    - In some cases spectral initialization and Frobenius decay can decrease accuracy unless used jointly (Table 1).

---

> > ### Comment · AnonReviewer2 · 2020-11-19
> > **Additional comments**
> >
> > Thanks the authors for the response.
> >
> > Below please find more details regarding to my previous comment on "It is well understood that spectral initialization is better than random initialization in low-rank literatures"
> >
> > Please see this survey paper on theoretical study on low-rank matrix factorization, Chapter 8 on spectral initialization.
> > https://arxiv.org/pdf/1809.09573.pdf
> >
> > This paper studies low-rank tensor completion, see Chapter 2 on the initialization: Section 2.1 discussed their spectral initialization procedure and Section 2.3 discussed why random initialization could not lead to good theoretical property.
> > https://arxiv.org/pdf/1911.04436.pdf
> >
> > Another earlier paper on the theoretical studies of low-rank tensor model, which also used spectral initialization.
> > https://jmlr.org/papers/volume15/anandkumar14b/anandkumar14b.pdf

---

> > > ### Author Response · Authors · 2020-11-20
> > > **Response to additional comments**
> > >
> > > Thank you for sharing these references, which study iterative methods for matrix factorization/tensor completion. These papers do not study initialization in the context of deep learning, which is the focus of our work, so their results are not directly applicable to our nonlinear setting. Another difference is their use of data-dependent initializations, with the goal of starting close to the quantity being approximated; this closeness is used to provide guarantees for gradient descent. In contrast we decompose a random initialization, so we cannot be approximating a data-dependent target and must be improving performance for some other reason, such as the one we propose. We thus maintain that our comparison of initialization techniques for factorized neural layers is novel and the results do not follow from prior work.
> > >
> > > We have uploaded a revision that includes the first two papers in the related work section as useful examples outside deep learning where spectral methods provide improved initializations for gradient descent. Note that the last reference studies the decomposition problem itself, and so cannot initialize using a tensor decomposition (in fact they initialize their Algorithm 1 using the uniform distribution over the unit sphere).

---

> > > > ### Comment · AnonReviewer2 · 2020-11-24
> > > > **I raise the score**
> > > >
> > > > Thank the authors for these additional clarification. I think these additional explanations relieve my former concern. So I increased score to 6.

---

### Author Response · Authors · 2020-11-16
**General response and revision notes**

Thank you to the reviewers for their feedback. We first address two concerns and then summarize changes made in the revised draft. Note that in addition to the revision we have also uploaded code as supplementary material.

### General author response:
1. _Novelty of the contributions_: while the two schemes we study are straightforward, to our knowledge they have not been clearly formalized, especially for training factorized networks from scratch. We believe their simplicity, the significant improvements we achieve on several tasks, and the theoretical foundations we provide will lead to more widespread study and use in future work. This is summarized in the following contributions of our paper:
   - A formalization of factorized neural layers that covers not only known applications such as low-rank networks but also new settings such as self-taught knowledge distillation and multi-head attention. To our knowledge the latter two have not been previously considered as part of the same framework.
   - The first mathematical and experimental study of spectral initialization and Frobenius decay, yielding an explanation for why they might improve upon standard approaches. This study also yields the new insight that regular weight-decay implicitly penalizes the nuclear norm of the product.
   - The first extensive comparison of low-rank and sparsity-based methods for low-memory model training, showing the surprising result that the former is superior in certain settings.
   - To our knowledge, the first application of these techniques to tensor-factorization approaches, demonstrating their usefulness.
   - In our estimation, the most convincing existing results for self-taught knowledge distillation. Previous published work in this area (Arora et al., 2018; Guo et al., 2020) considers only networks with accuracies well below ResNet performance on CIFAR-10/100. The absolute improvements (+1% on CIFAR-10/+2% on CIFAR-100) are also much better on ResNet56 than the latest unpublished work (Cao et al., 2020), although direct comparison is difficult because it is unclear what architecture variant they used.
   - An improved regularization scheme for multi-head attention and a demonstration of how to apply Frobenius decay to “decoupled weight-decay” optimizers (Loshchilov & Hutter, 2019).
2. _Simpler, SVD-free alternatives to spectral initialization_: we agree that it would be interesting to study alternative initializations, with the caveat that in the low-rank case one cannot obtain a factorization whose product is i.i.d. Gaussian without sacrificing independence, since an i.i.d. Gaussian ensemble is full-rank with probability 1. We thus must find other reasonable criteria, such as the approximation guarantee of SVD. Furthermore, we believe that spectral initialization can itself be viewed as a simple approach, as it is hyperparameter-free and can be computed on a CPU. In our experiments the single SVD step is a negligible fraction of the total training time, including on fairly large models such as BERT-Large and WideResNet28-10.

### Changes in the revision:
1. Additional plots (Figure 1, right; Figure 2, center and right) to address experimental questions by AnonReviewer3 and AnonReviewer4.
2. Additional discussion in Section 2 to address clarity questions by AnonReviewer1 and AnonReviewer4.
3. Conclusion (Section 7) to discuss points raised by AnonReviewer3.
4. Changes to Table 2 numbers to correct the results for the unfactorized WideResNet28-10 baseline, originally computed without weight-decay.
5. Changes to Table 3 numbers to report final test accuracy rather than best test accuracy; differences between methods are largely preserved.
6. Minor fixes and additional details added to the Appendix.

---

### Author Response · Authors · 2022-10-07
**Errata**

The [arXiv version of this work](https://arxiv.org/abs/2105.01029) has been amended following an error in the discussion pointed out by [Kamalakara et al. (2022)](https://arxiv.org/abs/2209.13569v1).

---

### Decision · Program_Chairs · 2021-01-07
**Final Decision**

**Decision:**

Accept (Poster)

**Comment:**

This paper applies spectral initialization and weight decay to neural nets with factorized layers. Although these ideas have been extensively studied in other areas, formalizing and applying them to deep neural nets is of potential interest to the community. The simulation results are nice, especially the experiments on compression methods (comparison to sparse pruning e.g. lottery tickets) and Transformers. I recommend acceptance.